# A Lightweight Method for Tackling Unknown Participation Statistics in Federated Averaging

**Shiqiang Wang**
IBM T. J. Watson Research Center
Yorktown Heights, NY 10598
wangshiq@us.ibm.com

**Mingyue Ji**
Department of ECE, University of Utah
Salt Lake City, UT 84112
mingyue.ji@utah.edu

## Abstract

In federated learning (FL), clients usually have diverse participation statistics that are unknown a priori, which can significantly harm the performance of FL if not handled properly. Existing works aiming at addressing this problem are usually based on global variance reduction, which requires a substantial amount of additional memory in a multiplicative factor equal to the total number of clients. An important open problem is to find a lightweight method for FL in the presence of clients with unknown participation rates. In this paper, we address this problem by *adapting the aggregation weights* in federated averaging (FedAvg) based on the participation history of each client. We first show that, with heterogeneous participation statistics, FedAvg with non-optimal aggregation weights can diverge from the optimal solution of the original FL objective, indicating the need of finding optimal aggregation weights. However, it is difficult to compute the optimal weights when the participation statistics are unknown. To address this problem, we present a new algorithm called FedAU, which improves FedAvg by adaptively weighting the client updates based on online estimates of the optimal weights without knowing the statistics of client participation. We provide a theoretical convergence analysis of FedAU using a novel methodology to connect the estimation error and convergence. Our theoretical results reveal important and interesting insights, while showing that FedAU converges to an optimal solution of the original objective and has desirable properties such as linear speedup. Our experimental results also verify the advantage of FedAU over baseline methods with various participation patterns.

## 1 Introduction

We consider the problem of finding $\mathbf{x} \in \mathbb{R}^d$ that minimizes the distributed finite-sum objective:

$$f(\mathbf{x}) := \frac{1}{N} \sum_{n=1}^{N} F_n(\mathbf{x}), \tag{1}$$

where each individual (local) objective $F_n(\mathbf{x})$ is only computable at the client $n$. This problem often arises in the context of federated learning (FL) (Kairouz et al., 2021; Li et al., 2020a; Yang et al., 2019), where $F_n(\mathbf{x})$ is defined on client $n$'s local dataset, $f(\mathbf{x})$ is the global objective, and $\mathbf{x}$ is the parameter vector of the model being trained. Each client keeps its local dataset to itself, which is not shared with other clients or the server. It is possible to extend (1) to weighted average with positive coefficients multiplied to each $F_n(\mathbf{x})$, but for simplicity, we consider such coefficients to be included in $\{F_n(\mathbf{x}) : \forall n\}$ (see Appendix A.1) and do not write them out.

Federated averaging (FedAvg) is a commonly used algorithm for minimizing (1), which alternates between *local updates* at each client and *parameter aggregation* among multiple clients with the help of a server (McMahan et al., 2017). However, there are several challenges in FedAvg, including data heterogeneity and partial participation of clients, which can cause performance degradation and even non-convergence if the FedAvg algorithm is improperly configured.

**Unknown, Uncontrollable, and Heterogeneous Participation of Clients.** Most existing works on FL with partial client participation assume that the clients participate according to a known or controllable random process (Chen et al., 2022; Fraboni et al., 2021a; Karimireddy et al., 2020; Li et al., 2020b;c; Yang et al., 2021). In practice, however, it is common for clients to have heterogeneous

and time-varying computation power and network bandwidth, which depend on both the inherent characteristics of each client and other tasks that concurrently run in the system. This generally leads to *heterogeneous participation statistics* across clients, which are *difficult to know a priori* due to their complex dependency on various factors in the system (Wang et al., 2021). It is also generally *impossible to fully control* the participation statistics, due to the randomness of whether a client can successfully complete a round of model updates (Bonawitz et al., 2019).

The problem of having heterogeneous and unknown participation statistics is that it may cause the result of FL to be biased towards certain local objectives, which *diverges* from the optimum of the original objective in (1). In FL, data heterogeneity across clients is a common phenomenon, resulting in diverse local objectives $\{F_n(\mathbf{x})\}$. The participation heterogeneity is often correlated with data heterogeneity, because the characteristics of different user populations may be correlated with how powerful their devices are. Intuitively, when some clients participate more frequently than others, the final FL result will be benefiting the local objectives of those frequently participating clients, causing a possible discrimination for clients that participate less frequently.

A few recent works aiming at addressing this problem are based on the idea of global variance reduction by saving the most recent updates of all the clients, which requires a substantial amount of additional memory in the order of $Nd$, i.e., the total number of clients times the dimension of the model parameter vector (Gu et al., 2021; Jhunjhunwala et al., 2022; Yan et al., 2020; Yang et al., 2022). This additional memory consumption is either incurred at the server or evenly distributed to all the clients. For practical FL systems with many clients, this causes unnecessary memory usage that affects the overall capability and performance of the system. Therefore, we ask the following important question in this paper:

*Is there a **lightweight** method that **provably** minimizes the original objective in (1), when the participation statistics of clients are unknown, uncontrollable, and heterogeneous?*

We leverage the insight that we can apply different weights to different clients' updates in the parameter aggregation stage of FedAvg. If this is done properly, the effect of heterogeneous participation can be canceled out so that we can minimize (1), as shown in existing works that assume known participation statistics (Chen et al., 2022; Fraboni et al., 2021a; Li et al., 2020b;c). However, in our setting, we *do not know* the participation statistics a priori, which makes it *challenging to compute (estimate) the optimal aggregation weights*. It is also non-trivial to quantify the impact of estimation error on convergence.

**Our Contributions.** We thoroughly analyze this problem and make the following novel contributions.

1. To motivate the need for adaptive weighting in parameter aggregation, we show that FedAvg with non-optimal weights minimizes a different objective (defined in (2)) instead of (1).

2. We propose a *lightweight procedure for estimating the optimal aggregation weight* at each client $n$ as part of the overall FL process, based on client $n$'s participation history. We name this new algorithm FedAU, which stands for FedAvg with adaptive weighting to support unknown participation statistics.

3. We analyze the convergence upper bound of FedAU, using a novel method that first obtains a *weight error term* in the convergence bound and then further bounds the weight error term via a bias-variance decomposition approach. Our result shows that *FedAU converges to an optimal solution of the original objective* (1). In addition, a desirable *linear speedup* of convergence with respect to the number of clients is achieved when the number of FL rounds is large enough.

4. We verify the advantage of FedAU in experiments with several datasets and baselines, with a *variety of participation patterns* including those that are independent, Markovian, and cyclic.

**Related Work.** Earlier works on FedAvg considered the convergence analysis with full client participation (Gorbunov et al., 2021; Haddadpour et al., 2019; Lin et al., 2020; Malinovsky et al., 2023; Stich, 2019; Wang & Joshi, 2019; 2021; Yu et al., 2019), which do not capture the fact that only a subset of clients participates in each round in practical FL systems. Recently, partial client participation has came to attention. Some works analyzed the convergence of FedAvg where the statistics or patterns of client participation are known or controllable (Chen et al., 2022; Cho et al., 2023; Fraboni et al., 2021a;b; Karimireddy et al., 2020; Li et al., 2020b;c; Rizk et al., 2022; Wang & Ji, 2022; Yang et al., 2021). However, as pointed out by Bonawitz et al. (2019); Wang et al. (2021), the participation of clients in FL can have complex dependencies on the underlying system characteristics,

which makes it difficult to know or control each client's behavior a priori. A recent work analyzed the convergence for a re-weighted objective (Patel et al., 2022), where the re-weighting is essentially arbitrary for unknown participation distributions. Some recent works (Gu et al., 2021; Jhunjhunwala et al., 2022; Yan et al., 2020; Yang et al., 2022) aimed at addressing this problem using variance reduction, by including the most recent local update of each client in the global update, even if they do not participate in the current round. These methods require a substantial amount of additional memory to store the clients' local updates. In contrast, our work focuses on developing a lightweight algorithm that has virtually the same memory requirement as the standard FedAvg algorithm.

A related area is adaptive FL algorithms, where adaptive gradients (Reddi et al., 2021; Wang et al., 2022b;c) and adaptive local updates (Ruan et al., 2021; Wang et al., 2020) were studied. Some recent works viewed the adaptation of aggregation weights from different perspectives (Tan et al., 2022; Wang et al., 2022a; Wu & Wang, 2021), which do not address the problem of unknown participation statistics. All these methods are orthogonal to our work and can potentially work together with our algorithm. To the best of our knowledge, no prior work has studied weight adaptation in the presence of unknown participation statistics with provable convergence guarantees.

A uniqueness in our problem is that the statistics related to participation need to be collected across multiple FL rounds. Although Wang & Ji (2022) aimed at extracting a participation-specific term in the convergence bound, that approach still requires the aggregation weights in each round to sum to one (thus coordinated participation); it also requires an amplification procedure over multiple rounds for the bound to hold, making it difficult to tune the hyperparameters. In contrast, this paper considers uncontrolled and uncoordinated participation without sophisticated amplification mechanisms.

## 2 FedAvg with Pluggable Aggregation Weights

We begin by describing a generic FedAvg algorithm that includes a separate oracle for computing the aggregation weights, as shown in Algorithm 1. In this algorithm, there are a total of $T$ rounds, where each round $t$ includes $I$ steps of local stochastic gradient descent (SGD) at a participating client. For simplicity, we consider $I$ to be the same for all the clients, while noting that our algorithm and results can be extended to more general cases. We use $\gamma > 0$ and $\eta > 0$ to denote the local and global step sizes, respectively. The variable $\mathbf{x}_0$ is the initial model parameter, $\mathbb{1}_t^n$ is an identity function that is equal to one if client $n$ participates in round $t$ and zero otherwise, and $\mathbf{g}_n(\cdot)$ is the stochastic gradient of the local objective $F_n(\cdot)$ for each client $n$.

---

**Algorithm 1:** FedAvg with pluggable aggregation weights

---

**Input:** $\gamma, \eta, \mathbf{x}_0, I$; **Output:** $\{\mathbf{x}_t : \forall t\}$;

1  Initialize $t_0 \leftarrow 0, \mathbf{u} \leftarrow \mathbf{0}$;
2  **for** $t = 0, \ldots, T-1$ **do**
3     **for** $n = 1, \ldots, N$ *in parallel* **do**
4        Sample $\mathbb{1}_t^n$ from an *unknown* process;
5        **if** $\mathbb{1}_t^n = 1$ **then**
6           $\mathbf{y}_{t,0}^n \leftarrow \mathbf{x}_t$;
7           **for** $i = 0, \ldots, I-1$ **do**
8              $\mathbf{y}_{t,i+1}^n \leftarrow \mathbf{y}_{t,i}^n - \gamma \mathbf{g}_n(\mathbf{y}_{t,i}^n)$;
9           $\Delta_t^n \leftarrow \mathbf{y}_{t,I}^n - \mathbf{x}_t$;
10       **else**
11          $\Delta_t^n \leftarrow \mathbf{0}$;
12       $\omega_t^n \leftarrow \texttt{ComputeWeight}(\{\mathbb{1}_\tau^n : \tau < t\})$;
13    $\mathbf{x}_{t+1} \leftarrow \mathbf{x}_t + \frac{\eta}{N} \sum_{n=1}^N \omega_t^n \Delta_t^n$;

---

The main steps of Algorithm 1 are similar to those of standard FedAvg, but with a few notable items as follows. *1)* In Line 4, we clearly state that we do not have prior knowledge of the sampling process of client participation. *2)* Line 12 calls a separate oracle to compute the aggregation weight $\omega_t^n$ ($\omega_t^n > 0$) for client $n$ in round $t$. This computation is done on each client $n$ alone, without coordinating with other clients. We *do not* need to save the full sequence of participation record $\{\mathbb{1}_\tau^n : \tau < t\}$, because it is sufficient to save an aggregated metric of the participation record for weight computation. In Section 3, we will see that we use the average participation interval for weight computation in FedAU, where the average can be computed in an online manner. We also note that we do not include $\mathbb{1}_t^n$ in the current round $t$ for computing the weight, which is needed for the convergence analysis so that $\omega_t^n$ is independent of the local parameter $\mathbf{y}_{t,i}^n$ when the initial parameter of round $t$ (i.e., $\mathbf{x}_t$) is given. *3)* The parameter aggregation is weighted by $\omega_t^n$ for each client $n$ in Line 13.

**Objective Inconsistency with Improper Aggregation Weights.** We first show that without weight adaptation, FedAvg minimizes an alternative objective that is generally different from (1).

**Theorem 1** (Objective minimized at convergence, informal). *When $\mathbb{1}_t^n \sim \text{Bernoulli}(p_n)$ and the weights are time-constant, i.e., $\omega_t^n = \omega_n$ but generally $\omega_n$ may not be equal to $\omega_{n'}$ ($n \neq n'$), with properly chosen learning rates $\gamma$ and $\eta$ and some other assumptions, Algorithm 1 minimizes the following objective:*

$$h(\mathbf{x}) := \frac{1}{P} \sum_{n=1}^N \omega_n p_n F_n(\mathbf{x}), \tag{2}$$

*where $P := \sum_{n=1}^N \omega_n p_n$.*

A formal version of the theorem is given in Appendix B.4. Theorem 1 shows that, even in the special case where each client $n$ participates according to a Bernoulli distribution with probability $p_n$, choosing a constant aggregation weight such as $\omega_n = 1, \forall n$ as in standard FedAvg causes the algorithm to converge to a different objective that is weighted by $p_n$. As mentioned earlier, this implicit weighting discriminates clients that participate less frequently. In addition, since the participation statistics (here, the probabilities $\{p_n\}$) of clients are unknown, the exact objective being minimized is also unknown, and it is generally unreasonable to minimize an unknown objective. This means that *it is important to design an adaptive method to find the aggregation weights*, so that we can minimize (1) even when the participation statistics are unknown, which is our focus in this paper.

*The full proofs of all mathematical claims are in Appendix B.*

## 3 FedAU: Estimation of Optimal Aggregation Weights

In this section, we describe the computation of aggregation weights $\{\omega_t^n\}$ based on the participation history observed at each client, which is the core of our FedAU algorithm that extends FedAvg. Our goal is to choose $\{\omega_t^n\}$ to minimize the original objective (1) as close as possible.

**Intuition.** We build from the intuition in Theorem 1 and design an aggregation weight adaptation algorithm that works for *general participation patterns*, i.e., not limited to the Bernoulli distribution considered in Theorem 1. From (2), we see that if we can choose $\omega_n = 1/p_n$, the objective being minimized is the same as (1). We note that $p_n \approx \frac{1}{T} \sum_{t=0}^{T-1} \mathbb{1}_t^n$ for each client $n$ when $T$ is large, due to ergodicity of the Bernoulli distribution considered in Theorem 1. Extending to general participation patterns that are not limited to the Bernoulli distribution, intuitively, we would like to choose the weight $\omega_n$ to be inversely proportional to the average frequency of participation. In this way, the bias caused by lower participation frequency is "canceled out" by the higher weight used in aggregation. Based on this intuition, our goal of aggregation weight estimation is as follows.

**Problem 1** (Goal of Weight Estimation, informal). *Choose $\{\omega_t^n\}$ so that its long-term average (i.e., for large $T$) $\frac{1}{T} \sum_{t=0}^{T-1} \omega_t^n$ is close to $\frac{1}{\frac{1}{T} \sum_{t=0}^{T-1} \mathbb{1}_t^n}$, for each $n$.*

Some previous works have discovered this need of debiasing the skewness of client participation (Li et al., 2020c; Perazzone et al., 2022) or designing the client sampling scheme to ensure that the updates are unbiased (Fraboni et al., 2021a; Li et al., 2020b). However, in our work, we consider the more realistic case where the participation statistics are unknown, uncontrollable, and heterogeneous. In this case, we *are unable to directly find* the optimal aggregation weights because we do not know the participation statistics a priori.

**Technical Challenge.** If we were to know the participation pattern for all the $T$ rounds, an immediate solution to Problem 1 is to choose $\omega_t^n$ (for each client $n$) to be equal to $T$ divided by the number of rounds where client $n$ participates. We can see that this solution is equal to the average *interval* between every two adjacent participating rounds, assuming that the first interval starts right before the first round $t = 0$. However, since we do not know the future participation pattern or statistics in each round $t$, we cannot directly apply this solution. In other words, in every round $t$, we need to perform an *online* estimation of the weight $\omega_t^n$ based on the participation history up to round $t - 1$.

A challenge in this online setting is that the estimation accuracy is related to the number of times each client $n$ has participated until round $t - 1$. When $t$ is small and client $n$ has not yet participated in any of the preceding rounds, we do not have any information about how to choose $\omega_t^n$. For an intermediate value of $t$ where client $n$ has participated only in a few rounds, we have limited information about the choice of $\omega_t^n$. In this case, if we directly use the average participation interval up to the $(t - 1)$-th round, the resulting $\omega_t^n$ can be far from its optimal value, i.e., the estimation has a high variance if

the client participation follows a random process. This is problematic especially when there exists a long interval between two rounds (both before the $(t-1)$-th round) where the client participates. Although the probability of the occurrence of such a long interval is usually low, when it occurs, it results in a long average interval for the first $t$ rounds when $t$ is relatively small, and using this long average interval as the value of $\omega_t^n$ may cause instability to the training process.

**Key Idea.** To overcome this challenge, we define a positive integer $K$ as a "cutoff" interval length. If a client has not participated for $K$ rounds, we consider $K$ to be a participation interval that we sample and start a new interval thereafter. In this way, we can limit the length of each interval by adjusting $K$. By setting $\omega_t^n$ to be the average of this possibly cutoff participation interval, we overcome the aforementioned challenge. From a theoretical perspective, we note that $\omega_t^n$ will be a biased estimation when $K < \infty$ and the bias will be larger when $K$ is smaller. In contrast, a smaller $K$ leads to a smaller variance of $\omega_t^n$, because we collect more samples in the computation of $\omega_t^n$ with a smaller $K$. Therefore, an insight here is that $K$ controls the *bias-variance tradeoff*[1] of the aggregation weight $\omega_t^n$. In Section 4, we will formally show this property and obtain desirable convergence properties of the weight error term and the overall objective function (1), by properly choosing $K$ in the theoretical analysis. Our experimental results in Section 5 also confirm that choosing an appropriate value of $K < \infty$ improves the performance in most cases.

**Online Algorithm.** Based on the above insight, we describe the procedure of computing the aggregation weights $\{\omega_t^n\}$, as shown in Algorithm 2. The computation is independent for each client $n$. In this algorithm, the variable $M_n$ denotes the number of (possibly cutoff) participation intervals that have been collected, and $S_n^\diamond$ denotes the the length of the last interval that is being computed. We compute the interval by incrementing $S_n^\diamond$ by one in every round, until the condition in Line 5 holds. When this condition holds, $S_n = S_n^\diamond$ is the actual length of the latest participation interval with possible cutoff. As explained above, we always start a new interval when $S_n^\diamond$ reaches $K$. Also note that we consider $\mathbb{1}_{t-1}^n$ instead of $\mathbb{1}_t^n$ in this condition and start

---

**Algorithm 2:** Weight computation in FedAU

**Input:** $K$, $\{\mathbb{1}_t^n : \forall t, n\}$; **Output:** $\{\omega_t^n : \forall t, n\}$;

1  **for** $n = 1, \dots, N$ *in parallel* **do**
2   Initialize $M_n \leftarrow 0$, $S_n^\diamond \leftarrow 0$, $\omega_0^n \leftarrow 1$;
3   **for** $t = 1, \dots, T-1$ **do**
4    $S_n^\diamond \leftarrow S_n^\diamond + 1$;
5    **if** $\mathbb{1}_{t-1}^n = 1$ *or* $S_n^\diamond = K$ **then**
6     $S_n \leftarrow S_n^\diamond$; // final interval computed
7     $\omega_t^n \leftarrow \begin{cases} S_n, & \text{if } M_n = 0 \\ \frac{M_n \cdot \omega_{t-1}^n + S_n}{M_n + 1}, & \text{if } M_n \geq 1 \end{cases}$;
8     $M_n \leftarrow M_n + 1$;
9     $S_n^\diamond \leftarrow 0$;
10   **else**
11    $\omega_t^n \leftarrow \omega_{t-1}^n$;

---

the loop from $t = 1$ in Line 3, to align with the requirement in Algorithm 1 that the weights are computed from the participation records before (not including) the current round $t$. For $t = 0$, we always use $\omega_t^n = 1$. In Line 7, we compute the weight using an online averaging method, which is equivalent to averaging over all the participation intervals that have been observed until each round $t$. With this method, we do not need to save all the previous participation intervals. Essentially, the computation in each round $t$ only requires three state variables that are scalars, including $M_n$, $S_n^\diamond$, and the previous round's weight $\omega_{t-1}^n$. This makes this algorithm *extremely memory efficient*.

In the full FedAU algorithm, we plug in the result of $\omega_n^t$ for each round $t$ obtained from Algorithm 2 into Line 12 of Algorithm 1. In other words, ComputeWeight in Algorithm 1 calls one step of update that includes Lines 4–11 of Algorithm 2.

**Compatibility with Privacy-Preserving Mechanisms.** In our FedAU algorithm, the aggregation weight computation (Algorithm 2) is done individually at each client, which only uses the client's participation states and does not use the training dataset or the model. When using these aggregation weights as part of FedAvg in Algorithm 1, the weight $\omega_t^n$ can be multiplied with the parameter update $\Delta_t^n$ at each client $n$ (and in each round $t$) *before* the update is transmitted to the server. In this way, methods such as secure aggregation (Bonawitz et al., 2017) can be applied directly, since the server only needs to compute a sum of the participating clients' updates. Differentially private FedAvg methods (Andrew et al., 2021; McMahan et al., 2018) can be applied in a similar way.

---

[1]Note that we focus on the aggregation weights here, which is different from classical concept of the bias-variance tradeoff of the model.

**Practical Implementation.** We will see from our experimental results in Section 5 that a coarsely chosen value of $K$ gives a reasonably good performance in practice, which means that we do not need to fine-tune $K$. There are also other engineering tweaks that can be made in practice, such as using an exponentially weighted average in Line 7 of Algorithm 2 to put more emphasis on the recent participation characteristics of clients. In an extreme case where each client participates only once, a possible solution is to group clients that have similar computation power (e.g., same brand/model of devices) and are in similar geographical locations together. They may share the same state variables $M_n$, $S_n^\diamond$, and $\omega_{t-1}^n$ used for weight computation in Algorithm 2. We note that according to the lower bound derived by Yang et al. (2022), if each client participates only once, it is impossible to have an algorithm to converge to the original objective without sharing additional information.

# 4 CONVERGENCE ANALYSIS

**Assumption 1.** *The local objective functions are L-smooth, such that*

$$\|\nabla F_n(\mathbf{x}) - \nabla F_n(\mathbf{y})\| \leq L \|\mathbf{x} - \mathbf{y}\|, \forall \mathbf{x}, \mathbf{y}, n. \tag{3}$$

**Assumption 2.** *The local stochastic gradients and unbiased with bounded variance, such that*

$$\mathbb{E}\left[\mathbf{g}_n(\mathbf{x})\mid \mathbf{x}\right] = \nabla F_n(\mathbf{x}) \text{ and } \mathbb{E}\left[\left\|\mathbf{g}_n(\mathbf{x}) - \nabla F_n(\mathbf{x})\right\|^2\Big| \mathbf{x}\right] \leq \sigma^2, \forall \mathbf{x}, n. \tag{4}$$

*In addition, the stochastic gradient noise $\mathbf{g}_n(\mathbf{x}) - \nabla F_n(\mathbf{x})$ is independent across different rounds (indexed by $t$), clients (indexed by $n$), and local update steps (indexed by $i$).*

**Assumption 3.** *The divergence between local and global gradients is bounded, such that*

$$\|\nabla F_n(\mathbf{x}) - \nabla f(\mathbf{x})\|^2 \leq \delta^2, \forall \mathbf{x}, n. \tag{5}$$

**Assumption 4.** *The client participation random variable $\mathbb{1}_t^n$ is independent across different $t$ and $n$. It is also independent of the stochastic gradient noise. For each client $n$, we define $p_n$ such that $\mathbb{E}[\mathbb{1}_t^n] = p_n$, i.e., $\mathbb{1}_t^n \sim \text{Bernoulli}(p_n)$, where the value of $p_n$ is unknown to the system a priori.*

Assumptions 1–3 are commonly used in the literature for the convergence analysis of FL algorithms (Cho et al., 2023; Wang & Ji, 2022; Yang et al., 2021). Our consideration of independent participation across clients in Assumption 4 is more realistic than the conventional setting of sampling among all the clients with or without replacement (Li et al., 2020c; Yang et al., 2021), because it is difficult to coordinate the participation across a large number of clients in practical FL systems.

**Challenge in Analyzing Time-Dependent Participation.** Regarding the assumption on the independence of $\mathbb{1}_t^n$ across time (round) $t$ in Assumption 4, the challenge in analyzing the more general time-dependent participation is due to the complex interplay between the randomness in stochastic gradient noise, participation identities $\{\mathbb{1}_t^n\}$, and estimated aggregation weights $\{\omega_t^n\}$. In particular, the first step in our proof of the general descent lemma (see Appendix B.3, the specific step is in (B.3.6)) would not hold if $\mathbb{1}_t^n$ is dependent on the past, because the past information is contained in $\mathbf{x}_t$ and $\{\omega_t^n\}$ that are conditions of the expectation. We emphasize that this is a *purely theoretical limitation*, and this time-independence of client participation has been assumed in the majority of works on FL with client sampling (Fraboni et al., 2021a;b; Karimireddy et al., 2020; Li et al., 2020b;c; Yang et al., 2021). The novelty in our analysis is that we consider the true values of $\{p_n\}$ to be *unknown* to the system. Our experimental results in Section 5 show that FedAU provides performance gains also for Markovian and cyclic participation patterns that are both time-dependent.

**Assumption 5.** *We assume that **either** of the following holds and define $\Psi_G$ accordingly.*

- **Option 1:** Nearly optimal weights. *Under the assumption that $\frac{1}{N}\sum_{n=1}^{N}(p_n\omega_t^n - 1)^2 \leq \frac{1}{81}$ for all $t$, we define $\Psi_G := 0$.*

- **Option 2:** Bounded global gradient. *Under the assumption that $\|\nabla f(\mathbf{x})\|^2 \leq G^2$ for any $\mathbf{x}$, we define $\Psi_G := G^2$.*

Assumption 5 is only needed for Theorem 2 (stated below) and not for Theorem 1. Here, the bounded global gradient assumption is a relaxed variant of the bounded stochastic gradient assumption commonly used in adaptive gradient algorithms (Reddi et al., 2021; Wang et al., 2022b;c). Although

focusing on very different problems, our FedAU method shares some similarities with adaptive gradient methods in the sense that we both adapt the weights used in model updates, where the adaptation is dependent on some parameters that progressively change during the training process; see Appendix A.2 for some further discussion. For the nearly optimal weights assumption, we can see that it holds if $8/9p_n \leq \omega_t^n \leq 10/9p_n$, which means a toleration of a relative error of $1/9 \approx 11\%$ from the optimal weight $1/p_n$. Theorem 2 holds under *either of* these two additional assumptions.

**Main Results.** We now present our main results, starting with the convergence of Algorithm 1 with arbitrary (but given) weights $\{\omega_n^t\}$ with respect to (w.r.t.) the original objective function in (1).

**Theorem 2** (Convergence error w.r.t. (1)). *Let* $\gamma \leq \frac{1}{4\sqrt{15}LI}$ *and* $\gamma\eta \leq \min\left\{\frac{1}{4LI}; \frac{N}{54LIQ}\right\}$, *where* $Q := \max_{t\in\{0,\ldots,T-1\}} \frac{1}{N}\sum_{n=1}^{N} p_n(\omega_t^n)^2$. *When Assumptions 1–5 hold, the result* $\{\mathbf{x}_t\}$ *obtained from Algorithm 1 satisfies:*

$$\frac{1}{T}\sum_{t=0}^{T-1} \mathbb{E}\left[\|\nabla f(\mathbf{x}_t)\|^2\right] \tag{6}$$

$$\leq \mathcal{O}\Bigg(\frac{\mathcal{F}}{\gamma\eta IT} + \frac{\Psi_G + \delta^2 + \gamma^2 L^2 I\sigma^2}{NT}\sum_{t=0}^{T-1}\sum_{n=1}^{N}\mathbb{E}\Big[(p_n\omega_t^n - 1)^2\Big] + \frac{\gamma\eta LQ(I\delta^2 + \sigma^2)}{N} + \gamma^2 L^2 I(I\delta^2 + \sigma^2)\Bigg),$$

*where* $\mathcal{F} := f(\mathbf{x}_0) - f^*$, *and* $f^* := \min_{\mathbf{x}} f(\mathbf{x})$ *is the truly minimum value of the objective in* (1).

The proof of Theorem 2 includes a novel step to obtain $\frac{1}{NT}\sum_{t=0}^{T-1}\sum_{n=1}^{N}\mathbb{E}\left[(p_n\omega_t^n - 1)^2\right]$ (ignoring the other constants), referred to as the *weight error term*, that characterizes how the aggregation weights $\{\omega_n^t\}$ affect the convergence. Next, we focus on $\{\omega_t^n\}$ obtained from Algorithm 2.

**Theorem 3** (Bounding the weight error term). *For* $\{\omega_t^n\}$ *obtained from Algorithm 2, when* $T \geq 2$,

$$\frac{1}{NT}\sum_{t=0}^{T-1}\sum_{n=1}^{N}\mathbb{E}\left[(p_n\omega_t^n - 1)^2\right] \leq \mathcal{O}\left(\frac{K\log T}{T} + \frac{1}{N}\sum_{n=1}^{N}(1-p_n)^{2K}\right). \tag{7}$$

The proof of Theorem 3 is based on analyzing the unique statistical properties of the possibly cutoff participation interval $S_n$ obtained in Algorithm 2. The first term of the bound in (7) is related to the variance of $\omega_t^n$. This term increases linearly in $K$, because when $K$ gets larger, the minimum number of samples of $S_n$ that are used for computing $\omega_t^n$ gets smaller, thus the variance upper bound becomes larger. The second term of the bound in (7) is related to the bias of $\omega_t^n$, which measures how far $\mathbb{E}\left[\omega_t^n\right]$ departs from the desired quantity of $1/p_n$. Since $0 < p_n \leq 1$, this term decreases exponentially in $K$. This result *confirms the bias-variance tradeoff* of $\omega_t^n$ that we mentioned earlier.

**Corollary 4** (Convergence of FedAU). *Let* $K = \lceil\log_c T\rceil$ *with* $c := 1/(1 - \min_n p_n)^2$, $\gamma = \min\left\{\frac{1}{LI\sqrt{T}}; \frac{1}{4\sqrt{15}LI}\right\}$, *and choose* $\eta$ *such that* $\gamma\eta = \min\left\{\sqrt{\frac{\mathcal{F}N}{Q(I\delta^2+\sigma^2)LIT}}; \frac{1}{4LI}; \frac{N}{54LIQ}\right\}$. *When* $T \geq 2$, *the result* $\{\mathbf{x}_t\}$ *obtained from Algorithm 1 that uses* $\{\omega_t^n\}$ *obtained from Algorithm 2 satisfies*

$$\frac{1}{T}\sum_{t=0}^{T-1}\mathbb{E}\left[\|\nabla f(\mathbf{x}_t)\|^2\right]$$

$$\leq \mathcal{O}\left(\frac{\sigma\sqrt{L\mathcal{F}Q}}{\sqrt{NIT}} + \frac{\delta\sqrt{L\mathcal{F}Q}}{\sqrt{NT}} + \frac{\left(\Psi_G + \delta^2 + \frac{\sigma^2}{IT}\right)R\log^2 T}{T} + \frac{L\mathcal{F}\left(1+\frac{Q}{N}\right) + \delta^2 + \frac{\sigma^2}{I}}{T}\right), \tag{8}$$

*where $Q$ and $\Psi_G$ are defined in Theorem 2 and* $R := 1/\log c$.

The result in Corollary 4 is the convergence upper bound of the full FedAU algorithm. Its proof involves further bounding (7) in Theorem 3, when choosing $K = \log_c T$, and plugging back the result along with the values of $\gamma$ and $\eta$ into Theorem 2. It shows that, with properly estimated aggregation weights $\{\omega_t^n\}$ using Algorithm 2, the error approaches zero as $T \to \infty$, although the actual participation statistics are unknown. The first two terms of the bound in (8) dominate when $T$ is large enough, which are related to the stochastic gradient variance $\sigma^2$ and gradient divergence $\delta^2$. The error caused by the fact that $\{p_n\}$ is unknown is captured by the third term of the bound in (8), which has an order of $\mathcal{O}(\log^2 T/T)$. We also see that, as long as we maintain $T$ to be large enough so that the first two terms of the bound in (8) dominate, we can achieve the desirable property of *linear speedup* in $N$. This means that we can keep the same convergence error by increasing the number of clients ($N$) and decreasing the number of rounds ($T$), to the extent that $T$ remains large enough. Our result also recovers existing convergence bounds for FedAvg in the case of known participation probabilities (Karimireddy et al., 2020; Yang et al., 2021); see Appendix A.3 for details.

## 5 EXPERIMENTS

We evaluate the performance of FedAU in experiments. *More experimental setup details, including the link to the code, and results are in Appendices C and D, respectively.*

**Datasets, Models, and System.** We consider four image classification tasks, with datasets including SVHN (Netzer et al., 2011), CIFAR-10 (Krizhevsky & Hinton, 2009), CIFAR-100 (Krizhevsky & Hinton, 2009), and CINIC-10 (Darlow et al., 2018), where CIFAR-100 has 100 classes (labels) while the other datasets have 10 classes. We use FL train convolutional neural network (CNN) models of slightly different architectures for these tasks. We simulate an FL system that includes a total of $N = 250$ clients, where each $n$ has its own participation pattern.

**Heterogeneity.** Similar to existing works (Hsu et al., 2019; Reddi et al., 2021), we use a Dirichlet distribution with parameter $\alpha_d = 0.1$ to generate the class distribution of each client's data, for a setup with non-IID data across clients. Here, $\alpha_d$ specifies the degree of *data heterogeneity*, where a smaller $\alpha_d$ indicates a more heterogeneous data distribution. In addition, to simulate the correlation between data distribution and client participation frequency as motivated in Section 1, we generate a class-wide participation probability distribution that follows a Dirichlet distribution with parameter $\alpha_p = 0.1$. Here, $\alpha_p$ specifies the degree of *participation heterogeneity*, where a smaller $\alpha_p$ indicates more heterogeneous participation across clients. We generate client participation patterns following a random process that is either Bernoulli (independent), Markovian, or cyclic, and study the performance of these types of participation patterns in different experiments. The participation patterns have a stationary probability $p_n$, for each client $n$, that is generated according to a combination of the two aforementioned Dirichlet distributions, and the details are explained in Appendix C.6. We enforce the minimum $p_n$, $\forall n$, to be $0.02$ in the main experiments, which is relaxed later. This generative approach creates an experimental scenario with non-IID client participation, while our FedAU algorithm and most baselines still do not know the actual participation statistics.

**Baselines.** We compare our FedAU algorithm with several baselines. The first set of baselines includes algorithms that compute an average of parameters over either all the participating clients (*average participating*) or all the clients (*average all*) in the aggregation stage of each round, where the latter case includes updates of non-participating clients that are equal to zero as part of averaging. These two baselines encompass most existing FedAvg implementations (e.g., McMahan et al. (2017); Patel et al. (2022); Yang et al. (2021)) that do not address the bias caused by heterogeneous participation statistics. They do not require additional memory or knowledge, thus they work under the same

Table 1: Accuracy results (in %) on training and test data

| Participation pattern | Dataset | SVHN | | CIFAR-10 | | CIFAR-100 | | CINIC-10 | |
|---|---|---|---|---|---|---|---|---|---|
| | Method / Metric | Train | Test | Train | Test | Train | Test | Train | Test |
| Bernoulli | FedAU (ours, $K \to \infty$) | 90.4±0.5 | 89.3±0.5 | 85.4±0.4 | 77.1±0.4 | 63.4±0.6 | **52.3**±0.4 | 65.2±0.5 | 61.5±0.4 |
| | FedAU (ours, $K = 50$) | **90.6**±0.4 | **89.6**±0.4 | **86.0**±0.5 | **77.3**±0.3 | **63.8**±0.3 | 52.1±0.6 | **66.7**±0.3 | **62.7**±0.2 |
| | Average participating | 89.1±0.3 | 87.2±0.3 | 83.5±0.9 | 74.1±0.8 | 59.3±0.4 | 48.8±0.7 | 61.1±2.3 | 56.6±2.0 |
| | Average all | 88.5±0.5 | 87.0±0.3 | 81.0±0.9 | 72.7±0.9 | 58.2±0.4 | 47.9±0.5 | 60.5±2.3 | 56.2±2.0 |
| | FedVarp (250× memory) | 89.6±0.5 | 88.9±0.5 | 84.2±0.3 | 77.9±0.2 | 57.2±0.9 | 49.2±0.8 | 64.4±0.6 | 62.0±0.5 |
| | MIFA (250× memory) | 89.4±0.3 | 88.7±0.2 | 83.5±0.6 | 77.5±0.3 | 55.8±1.1 | 48.4±0.7 | 63.8±0.7 | 61.5±0.5 |
| | Known participation statistics | 89.2±0.5 | 88.4±0.5 | 84.3±0.5 | 77.0±0.5 | 59.4±0.7 | 50.6±0.4 | 63.2±0.6 | 60.5±0.5 |
| Markovian | FedAU (ours, $K \to \infty$) | 90.5±0.4 | 89.3±0.4 | 85.3±0.3 | 77.1±0.3 | 63.2±0.5 | **51.8**±0.3 | 64.9±0.3 | 61.2±0.2 |
| | FedAU (ours, $K = 50$) | **90.6**±0.3 | **89.5**±0.3 | **85.9**±0.4 | **77.2**±0.3 | **63.5**±0.4 | 51.7±0.3 | **66.3**±0.4 | **62.3**±0.2 |
| | Average participating | 89.0±0.3 | 87.1±0.2 | 83.4±0.9 | 74.2±0.7 | 59.2±0.4 | 48.6±0.4 | 61.5±2.3 | 56.9±1.9 |
| | Average all | 88.4±0.6 | 86.8±0.7 | 80.8±1.0 | 72.5±0.5 | 57.8±0.9 | 47.7±0.5 | 59.9±2.8 | 55.7±2.2 |
| | FedVarp (250× memory) | 89.6±0.3 | 88.6±0.2 | 84.0±0.3 | 77.8±0.2 | 56.4±1.1 | 48.8±0.5 | 64.6±0.4 | 62.1±0.4 |
| | MIFA (250× memory) | 89.1±0.3 | 88.4±0.2 | 83.0±0.4 | 77.2±0.4 | 55.1±1.2 | 48.1±0.6 | 63.5±0.7 | 61.2±0.6 |
| | Known participation statistics | 89.5±0.2 | 88.6±0.2 | 84.5±0.4 | 76.9±0.3 | 59.7±0.5 | 50.3±0.5 | 63.5±0.9 | 60.7±0.6 |
| Cyclic | FedAU (ours, $K \to \infty$) | 89.8±0.6 | 88.7±0.6 | 84.2±0.8 | 76.3±0.7 | 60.9±0.6 | 50.6±0.3 | 63.5±1.0 | 60.0±0.8 |
| | FedAU (ours, $K = 50$) | **89.9**±0.6 | **88.8**±0.6 | **84.8**±0.6 | **76.6**±0.4 | **61.3**±0.8 | **51.0**±0.5 | **64.5**±0.9 | **60.9**±0.7 |
| | Average participating | 87.4±0.5 | 85.5±0.7 | 81.6±1.2 | 73.3±0.8 | 58.1±1.0 | 48.3±0.8 | 58.9±2.1 | 55.0±1.6 |
| | Average all | 89.1±0.8 | 87.4±0.8 | 83.1±1.0 | 73.8±0.8 | 59.7±0.3 | 48.8±0.4 | 62.9±1.7 | 57.6±1.5 |
| | FedVarp (250× memory) | 84.8±0.5 | 83.9±0.6 | 79.7±0.9 | 75.3±0.7 | 50.9±0.5 | 45.9±0.4 | 60.4±0.7 | 58.5±0.6 |
| | MIFA (250× memory) | 78.6±1.2 | 77.4±1.1 | 73.0±1.3 | 70.6±1.1 | 44.8±0.6 | 41.1±0.6 | 51.2±1.0 | 50.2±0.9 |
| | Known participation statistics | 89.9±0.7 | 88.7±0.6 | 83.6±0.7 | 76.1±0.5 | 60.2±0.4 | 50.8±0.4 | 62.6±0.8 | 59.8±0.7 |

*Note to the table.* The top part of the sub-table for each participation pattern includes our method and baselines in the same setting. The bottom part of each sub-table includes baselines that require either additional memory or known participation statistics. For each column, the best values in the top and bottom parts are highlighted with **bold** and underline, respectively. The total number of rounds is $2,000$ for SVHN; $10,000$ for CIFAR-10 and CINIC-10; $20,000$ for CIFAR-100. The mean and standard deviation values shown in the table are computed over experiments with 5 different random seeds, for the average accuracy over the last 200 rounds (measured at an interval of 10 rounds).

system assumptions as FedAU. The second set of baselines has algorithms that require *extra* resources or information, including *FedVarp* (Jhunjhunwala et al., 2022) and *MIFA* (Gu et al., 2021), which require $N = 250$ times of memory, and an idealized baseline that assumes *known participation statistics* and weighs the clients' contributions using the reciprocal of the stationary participation probability. For each baseline, we performed a separate grid search to find the best $\gamma$ and $\eta$.

**Results.** The main results are shown in Table 1, where we choose $K = 50$ for FedAU with finite $K$ based on a simple rule-of-thumb without detailed search. Our general observation is that *FedAU provides the highest accuracy compared to almost all the baselines*, including those that require additional memory and known participation statistics, except for the test accuracy on the CIFAR-10 dataset where FedVarp performs the best. Choosing $K = 50$ generally gives a better performance than choosing $K \to \infty$ for FedAU, which aligns with our discussion in Section 3.

The reason that FedAU can perform better than FedVarp and MIFA is that these baselines keep historical local updates, which may be outdated when some clients participate infrequently. Updating the global model parameter with outdated local updates can lead to slow convergence, which is similar to the consequence of having stale updates in asynchronous SGD (Recht et al., 2011). In contrast, at the beginning of each round, participating clients in FedAU always start with the latest global parameter obtained from the server. This avoids stale updates, and we compensate heterogeneous participation statistics by adapting the aggregation weights, which is a fundamentally different and more efficient method compared to tracking historical updates as in FedVarp and MIFA.

It is surprising that FedAU even performs better than the case with known participation statistics. To understand this phenomenon, we point out that in the case of Bernoulli-distributed participation with very low probability (e.g., $p_n = 0.02$), the empirical probability of a sample path of a client's participation can diverge significantly from $p_n$. For $T = 10,000$ rounds, the standard deviation of the total number of participated rounds is $\sigma' := \sqrt{Tp_n(1-p_n)} = 0.0196 = 14$ while the mean is $\mu' := Tp_n = 200$. Considering the range within $2\sigma'$, we know that the optimal participation weight when seen on the *empirical* probability ranges from $T/(\mu' + 2\sigma') \approx 43.9$ to $T/(\mu' - 2\sigma') \approx 58.1$, while the optimal weight computed on the *model-based* probability is $1/p_n = 50$. Our FedAU algorithm computes the aggregation weights from the *actual participation sample path* of each client, which captures the actual client behavior and empirically performs better than using $1/p_n$ even if $p_n$ is known. Some experimental results that further explain this phenomenon are in Appendix D.4.

As mentioned earlier, we lower-bounded $p_n$, $\forall n$, by 0.02 for the main results. Next, we consider different lower bounds of $p_n$, where a smaller lower bound of $p_n$ means that there exist clients that participate less frequently. The performance of FedAU with different choices of $K$ and different lower bounds of $p_n$ is shown in Figure 1. We observe that choosing $K = 50$ always gives the best performance; the performance remains similar even when the lower bound of $p_n$ is small and there exist some clients that participate very infrequently. However, choosing a large $K$ (e.g., $K \geq 500$) significantly deteriorates the performance when the lower bound of $p_n$ is small. This means that having a finite cutoff interval $K$ of an intermediate value (i.e., $K = 50$ in our experiments) for aggregation weight estimation, which is a uniqueness of FedAU, is essential especially when very infrequently participating clients exist.

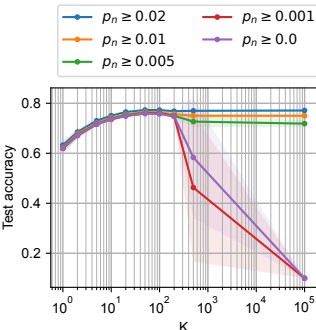

Figure 1: FedAU with different $K$ (CIFAR-10 with Bernoulli participation).

## 6  CONCLUSION

In this paper, we have studied the challenging practical FL scenario of having unknown participation statistics of clients. To address this problem, we have considered the adaptation of aggregation weights based on the participation history observed at each individual client. Using a new consideration of the bias-variance tradeoff of the aggregation weight, we have obtained the FedAU algorithm. Our analytical methodology includes a unique decomposition which yields a separate weight error term that is further bounded to obtain the convergence upper bound of FedAU. Experimental results have confirmed the advantage of FedAU with several client participation patterns. Future work can study the convergence analysis of FedAU with more general participation processes and the incorporation of aggregation weight adaptation into other types of FL algorithms.

## ACKNOWLEDGMENT

The work of M. Ji was supported by the National Science Foundation (NSF) CAREER Award 2145835.

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

# Appendix

## A  ADDITIONAL DISCUSSION

### A.1  EXTENDING OBJECTIVE (1) TO WEIGHTED AVERAGE

We note that our objective (1) can be easily extended to a weighted average of per-client empirical risk (i.e., average of sample losses), with arbitrary weights $\{q_n : \forall n\}$. To see this, let $\hat{F}_n(\mathbf{x})$ denote the (local) empirical risk of client $n$, and let $\Gamma := \sum_{n=1}^{N} q_n$. We can define the local objective of client $n$ as $F_n(\mathbf{x}) = \frac{q_n N}{\Gamma} \hat{F}_n(\mathbf{x})$, which gives us the global objective of

$$f(\mathbf{x}) = \frac{1}{N} \sum_{n=1}^{N} F_n(\mathbf{x}) = \frac{1}{\Gamma} \sum_{n=1}^{N} q_n \hat{F}_n(\mathbf{x}). \tag{A.1.1}$$

This objective is in a standard form seen in most FL papers. The extension allows us to give different importance to different clients, if needed. For simplicity, we do not write out the weights $\{q_n\}$ in the main paper, because this extension to arbitrary weights $\{q_n\}$ is straightforward, and such a simplification has also been made in various other works such as Jhunjhunwala et al. (2022); Karimireddy et al. (2020); Reddi et al. (2021); Wang & Ji (2022).

### A.2  ASSUMPTION ON BOUNDED GLOBAL GRADIENT

As stated in Theorem 2, our convergence result holds when *either* of the "bounded global gradient" assumption or the "nearly optimal weights" assumption holds. When the aggregation weights $\{\omega_t^n\}$ are nearly optimal satisfying $\frac{1}{N} \sum_{n=1}^{N} (p_n \omega_t^n - 1)^2 \leq \frac{1}{81}$, we do not need the bounded gradient assumption.

For the bounded gradient assumption itself, a stronger assumption of bounded stochastic gradient is used in related works on adaptive gradient algorithms (Reddi et al., 2021; Wang et al., 2022b;c), which implies an upper bound on the per-sample gradient. Compared to these works, we only require an upper bound on the global gradient, i.e., average of per-sample gradients, in our work. Although focusing on very different problems, our FedAU method shares some similarities with adaptive gradient methods in the sense that we both adapt the weights used in model updates, where the adaptation is dependent on some parameters that progressively change during the training process. The difference, however, is that our weight adaptation is based on each client's participation history, while adaptive gradient methods adapt the element-wise weights based on the historical model update vector. Nevertheless, the similarity in both methods leads to a technical (mathematical) step of bounding a "weight error" in the proofs, which is where the bounded gradient assumption is needed especially when the "weight error" itself cannot be bounded. In our work, this step is done in the proof of Theorem 2 (in Appendix B.5). In adaptive gradient methods, as an example, this step is on page 14 until Equation (4) in Reddi et al. (2021).

Again, we note that the bounded gradient assumption is only needed when the aggregation weights are estimated and the estimation error is large. This is seen in the two choices in Assumption 5; the convergence bound holds when either of these two conditions hold. Intuitively, this aligns with the reasoning of the need for bounding the "weight error".

### A.3  COMPARISON WITH EXISTING CONVERGENCE BOUNDS FOR FEDAVG

We compare our result in Corollary 4 with existing FedAvg convergence results, where the latter assumes known participation probabilities. Since most existing results consider equiprobable sampling of a certain number (denoted by $S$ here) of clients out of all the $N$ clients, we first convert our bound to the same setting so that it is comparable with existing results. We note that our convergence bound includes the parameter $Q$ that is defined as $Q := \max_{t \in \{0, \ldots, T-1\}} \frac{1}{N} \sum_{n=1}^{N} p_n (\omega_t^n)^2$ in Theorem 2. When we know the participation probabilities and choose $\omega_t^n = \frac{1}{p_n}$ for all $t$, we have $Q = \frac{1}{N} \sum_{n=1}^{N} \frac{1}{p_n}$. Further, for equiprobable sampling of $S$ clients out of a total of $N$ clients, we have $p_n = \frac{S}{N}$ and thus $Q = \frac{N}{S}$. Therefore, when $T$ is large and ignoring the other constants, our upper bound in Corollary 4 becomes $\mathcal{O}\left(\frac{\sqrt{Q}}{\sqrt{NT}}\right) = \mathcal{O}\left(\frac{1}{\sqrt{ST}}\right)$.

Considering existing results of FedAvg with partial participation where the probabilities are both homogeneous and known, Theorem 1 in Karimireddy et al. (2020) gives the same convergence bound of $\mathcal{O}\left(\frac{1}{\sqrt{ST}}\right)$ for non-convex objectives, and Corollary 2 in Yang et al. (2021) gives a covergence bound of $\mathcal{O}\left(\frac{\sqrt{I}}{\sqrt{ST}}\right)$. Here, we note that Karimireddy et al. (2020) express the bound on communication rounds while we give the bound on the square of gradient norm, but the two types of bounds are directly convertible to each other. Our bound of $\mathcal{O}\left(\frac{1}{\sqrt{ST}}\right)$ matches with Theorem 1 in Karimireddy et al. (2020) and improves over Corollary 2 in Yang et al. (2021). We also note that, in this special case, our result shows a *linear speedup* with respect to the number of participating clients, i.e., $S$, which is the same as the existing results in Karimireddy et al. (2020); Yang et al. (2021).

The uniqueness of our work compared to Karimireddy et al. (2020); Yang et al. (2021) and most other existing works is that we consider heterogeneous and unknown participation statistics (probabilities), where each client $n$ has its own participation probability $p_n$ that can be different from other clients. In contrast, Karimireddy et al. (2020); Yang et al. (2021) assume uniformly sampled clients where a fixed (and known) number of $S$ clients participate in each round. Our setup is more general where the number of clients that participate in each round can vary over time. Because of this generality, we cannot define a fixed value of $S$ in our convergence bound that holds for this general setup, so we use $Q$ to capture the statistical characteristics of client participation. When the overall probability distribution of client participation remains the same, increasing the total number of clients ($N$) has the same effect as increasing the number of participating clients ($S$), as we have shown above.

As a side note, when choosing $\omega_t^n = \frac{1}{p_n}$, the weight error term $\mathbb{E}[(p_n\omega_t^n - 1)^2]$ becomes zero and the third term in (8) in Corollary 4 will not exist, i.e., it becomes zero. See the proof in Appendix B.7 for why the third term in (8) is related to the weight error.

# B PROOFS

## B.1 PRELIMINARIES

We first note the following preliminary inequalities that we will use in the proofs without explaining them further.

We have

$$\left\| \frac{1}{M} \sum_{m=1}^{M} \mathbf{z}_m \right\|^2 \leq \frac{1}{M} \sum_{m=1}^{M} \|\mathbf{z}_m\|^2 \text{ and } \left\| \sum_{m=1}^{M} \mathbf{z}_m \right\|^2 \leq M \sum_{m=1}^{M} \|\mathbf{z}_m\|^2 \qquad \text{(B.1.1)}$$

for any $\mathbf{z}_m \in \mathbb{R}^d$ with $m \in \{1, 2, \ldots, M\}$, which is a direct consequence of Jensen's inequality.

We also have

$$\langle \mathbf{z}_1, \mathbf{z}_2 \rangle \leq \frac{\rho \|\mathbf{z}_1\|^2}{2} + \frac{\|\mathbf{z}_2\|^2}{2\rho}, \qquad \text{(B.1.2)}$$

for any $\mathbf{z}_1, \mathbf{z}_2 \in \mathbb{R}^d$ and $\rho > 0$, which is known as (the generalized version of) Young's inequality and also Peter-Paul inequality. A direct consequence of (B.1.2) is

$$\|\mathbf{z}_1 + \mathbf{z}_2\|^2 \leq (1 + b) \|\mathbf{z}_1\|^2 + \left(1 + \frac{1}{b}\right) \|\mathbf{z}_2\|^2, \qquad \text{(B.1.3)}$$

for some constant $b > 0$.

We also use the variance relation as follows:

$$\mathbb{E}\left[\|\mathbf{z}\|^2\right] = \|\mathbb{E}[\mathbf{z}]\|^2 + \mathbb{E}\left[\|\mathbf{z} - \mathbb{E}[\mathbf{z}]\|^2\right], \qquad \text{(B.1.4)}$$

for any $\mathbf{z} \in \mathbb{R}^d$, while noting that (B.1.4) also holds when all the expectations are conditioned on the same variable(s).

In addition, we use $\mathbb{E}_t[\cdot]$ to denote $\mathbb{E}[\cdot | \mathbf{x}_t, \{\omega_t^n\}]$ in short. We also assume that Assumptions 1–4 hold throughout our analysis.

## B.2 EQUIVALENT FORMULATION OF ALGORITHM 1

For the purpose of analysis, similar to Wang & Ji (2022), we consider an equivalent formulation of the original Algorithm 1, as shown in Algorithm B.2.1. In this algorithm, we assume that all the clients compute their local updates in Lines 5–8. This is logically equivalent to the practical setting where the clients that do not participate have no computation, because their computed update $\Delta_t^n$ has no effect in Line 10 if $\mathbb{1}_t^n = 0$, thus Algorithm 1 and Algorithm B.2.1 give the same output sequence $\{\mathbf{x}_t : \forall t\}$. Our proofs in the following sections consider the logically equivalent Algorithm B.2.1 for analysis and also use the notations defined in this algorithm.

---

**Algorithm B.2.1:** A logically equivalent version of Algorithm 1

**Input:** $\gamma, \eta, \mathbf{x}_0, I$
**Output:** $\{\mathbf{x}_t : \forall t\}$
1 Initialize $t_0 \leftarrow 0$, $\mathbf{u} \leftarrow \mathbf{0}$;
2 **for** $t = 0, \ldots, T - 1$ **do**
3     **for** $n = 1, \ldots, N$ *in parallel* **do**
4         Sample $\mathbb{1}_t^n$ from an *unknown* stochastic process;
5         $\mathbf{y}_{t,0}^n \leftarrow \mathbf{x}_t$;
6         **for** $i = 0, \ldots, I - 1$ **do**
7             $\mathbf{y}_{t,i+1}^n \leftarrow \mathbf{y}_{t,i}^n - \gamma \mathbf{g}_n(\mathbf{y}_{t,i}^n)$;   // In practice, no computation if $\mathbb{1}_t^n = 0$
8         $\Delta_t^n \leftarrow \mathbf{y}_{t,I}^n - \mathbf{x}_t$;
9         $\omega_t^n \leftarrow \texttt{ComputeWeight}(\{\mathbb{1}_\tau^n : \tau < t\})$;
10     $\mathbf{x}_{t+1} \leftarrow \mathbf{x}_t + \frac{\eta}{N} \sum_{n=1}^{N} \mathbb{1}_t^n \omega_t^n \Delta_t^n$;

---

### B.3 GENERAL DESCENT LEMMA

To prove the general descent lemma that is used to derive both Theorems 1 and 2, we first define the following generally weighted loss function.

**Definition B.3.1.** *Define*

$$\tilde{f}(\mathbf{x}) := \sum_{n=1}^{N} \varphi_n F_n(\mathbf{x}) \tag{B.3.1}$$

*where $\varphi_n \geq 0$ for all $n$ and $\sum_{n=1}^{N} \varphi_n = 1$.*

In (B.3.1), choosing $\varphi_n = \frac{1}{N}$ gives our original objective of $f(\mathbf{x})$. Note that we consider the updates in Algorithm B.2.1 to be still without weighting by $\varphi_n$, which allows us to quantify the convergence to a different objective when the aggregation weights are not properly chosen.

**Lemma B.3.1.** *Define $\tilde{\delta} := 2\delta$, we have*

$$\left\| \nabla F_n(\mathbf{x}) - \nabla \tilde{f}(\mathbf{x}) \right\|^2 \leq \tilde{\delta}^2, \forall \mathbf{x}, n. \tag{B.3.2}$$

*Proof.* From Assumption 3, we have

$$\left\| \nabla F_n(\mathbf{x}) - \nabla \tilde{f}(\mathbf{x}) \right\|^2 = \left\| \nabla F_n(\mathbf{x}) - \nabla f(\mathbf{x}) + \nabla f(\mathbf{x}) - \nabla \tilde{f}(\mathbf{x}) \right\|^2$$

$$\leq 2 \left\| \nabla F_n(\mathbf{x}) - \nabla f(\mathbf{x}) \right\|^2 + 2 \left\| \nabla f(\mathbf{x}) - \nabla \tilde{f}(\mathbf{x}) \right\|^2$$

$$= 2 \left\| \nabla F_n(\mathbf{x}) - \nabla f(\mathbf{x}) \right\|^2 + 2 \left\| \sum_{n=1}^{N} \varphi_n (\nabla f(\mathbf{x}) - \nabla F_n(\mathbf{x})) \right\|^2$$

$$\stackrel{(a)}{\leq} 2 \left\| \nabla F_n(\mathbf{x}) - \nabla f(\mathbf{x}) \right\|^2 + 2 \sum_{n=1}^{N} \varphi_n \left\| \nabla f(\mathbf{x}) - \nabla F_n(\mathbf{x}) \right\|^2$$

$$\leq 4\delta^2,$$

where we use the Jensen's inequality in (a). The final result follows due to $\tilde{\delta}^2 := 4\delta^2$. $\qquad\square$

**Lemma B.3.2.** *When $\gamma \leq \frac{1}{\sqrt{30}LI}$,*

$$\mathbb{E}_t \left[ \left\| \mathbf{y}_{t,i}^n - \mathbf{x}_t \right\|^2 \right] \leq 5I\gamma^2(\sigma^2 + 6I\tilde{\delta}^2) + 30I^2\gamma^2 \left\| \nabla \tilde{f}(\mathbf{x}_t) \right\|^2 \tag{B.3.3}$$

*Proof.* This lemma has the same form as in Yang et al. (2021, Lemma 2) and Reddi et al. (2021, Lemma 3), but we present it here for a single client $n$ instead of average over multiple clients.

For $i \in \{0, 1, 2, \dots, I-1\}$, we have

$$\mathbb{E}_t \left[ \left\| \mathbf{y}_{t,i+1}^n - \mathbf{x}_t \right\|^2 \right]$$

$$= \mathbb{E}_t \left[ \left\| \mathbf{y}_{t,i}^n - \mathbf{x}_t - \gamma \mathbf{g}_n(\mathbf{y}_{t,i}^n) \right\|^2 \right]$$

$$= \mathbb{E}_t \left[ \left\| \mathbf{y}_{t,i}^n - \mathbf{x}_t - \gamma \left( \mathbf{g}_n(\mathbf{y}_{t,i}^n) - \nabla F_n(\mathbf{y}_{t,i}^n) + \nabla F_n(\mathbf{y}_{t,i}^n) - \nabla F_n(\mathbf{x}_t) + \nabla F_n(\mathbf{x}_t) \right. \right.$$

$$\left. \left. - \nabla \tilde{f}(\mathbf{x}_t) + \nabla \tilde{f}(\mathbf{x}_t) \right) \right\|^2 \right]$$

$$\stackrel{(a)}{=} \mathbb{E}_t \left[ \left\| \gamma \left( \mathbf{g}_n(\mathbf{y}_{t,i}^n) - \nabla F_n(\mathbf{y}_{t,i}^n) \right) \right\|^2 \right]$$

$$+ 2\mathbb{E}_t \left[ \mathbb{E}_t \left[ \left\langle \gamma \left( \mathbf{g}_n(\mathbf{y}_{t,i}^n) - \nabla F_n(\mathbf{y}_{t,i}^n) \right), \mathbf{y}_{t,i}^n - \mathbf{x}_t - \gamma \left( \nabla F_n(\mathbf{y}_{t,i}^n) - \nabla F_n(\mathbf{x}_t) + \nabla F_n(\mathbf{x}_t) \right. \right. \right. \right.$$

$$\left. \left. \left. \left. - \nabla \tilde{f}(\mathbf{x}_t) + \nabla \tilde{f}(\mathbf{x}_t) \right) \right\rangle \Big| \mathbf{y}_{t,i}^n, \mathbf{x}_t \right] \right]$$

$$+ \mathbb{E}_t \left[ \left\| \mathbf{y}_{t,i}^n - \mathbf{x}_t - \gamma \left( \nabla F_n(\mathbf{y}_{t,i}^n) - \nabla F_n(\mathbf{x}_t) + \nabla F_n(\mathbf{x}_t) - \nabla \tilde{f}(\mathbf{x}_t) + \nabla \tilde{f}(\mathbf{x}_t) \right) \right\|^2 \right]$$

$$\overset{(b)}{=} \mathbb{E}_t \left[ \left\| \gamma \left( \mathbf{g}_n(\mathbf{y}_{t,i}^n) - \nabla F_n(\mathbf{y}_{t,i}^n) \right) \right\|^2 \right]$$

$$+ 2\mathbb{E}_t \left[ \left\langle \mathbb{E}_t \left[ \gamma \left( \mathbf{g}_n(\mathbf{y}_{t,i}^n) - \nabla F_n(\mathbf{y}_{t,i}^n) \right) \middle| \mathbf{y}_{t,i}^n, \mathbf{x}_t \right], \right. \right.$$

$$\left. \left. \mathbf{y}_{t,i}^n - \mathbf{x}_t - \gamma \left( \nabla F_n(\mathbf{y}_{t,i}^n) - \nabla F_n(\mathbf{x}_t) + \nabla F_n(\mathbf{x}_t) - \nabla \tilde{f}(\mathbf{x}_t) + \nabla \tilde{f}(\mathbf{x}_t) \right) \right\rangle \right]$$

$$+ \mathbb{E}_t \left[ \left\| \mathbf{y}_{t,i}^n - \mathbf{x}_t - \gamma \left( \nabla F_n(\mathbf{y}_{t,i}^n) - \nabla F_n(\mathbf{x}_t) + \nabla F_n(\mathbf{x}_t) - \nabla \tilde{f}(\mathbf{x}_t) + \nabla \tilde{f}(\mathbf{x}_t) \right) \right\|^2 \right]$$

$$\overset{(c)}{=} \mathbb{E}_t \left[ \left\| \gamma \left( \mathbf{g}_n(\mathbf{y}_{t,i}^n) - \nabla F_n(\mathbf{y}_{t,i}^n) \right) \right\|^2 \right]$$

$$+ \mathbb{E}_t \left[ \left\| \mathbf{y}_{t,i}^n - \mathbf{x}_t - \gamma \left( \nabla F_n(\mathbf{y}_{t,i}^n) - \nabla F_n(\mathbf{x}_t) + \nabla F_n(\mathbf{x}_t) - \nabla \tilde{f}(\mathbf{x}_t) + \nabla \tilde{f}(\mathbf{x}_t) \right) \right\|^2 \right]$$

$$\overset{(d)}{\leq} \mathbb{E}_t \left[ \left\| \gamma \left( \mathbf{g}_n(\mathbf{y}_{t,i}^n) - \nabla F_n(\mathbf{y}_{t,i}^n) \right) \right\|^2 \right] + \left( 1 + \frac{1}{2I - 1} \right) \mathbb{E}_t \left[ \left\| \mathbf{y}_{t,i}^n - \mathbf{x}_t \right\|^2 \right]$$

$$+ 2I \mathbb{E}_t \left[ \left\| \gamma \left( \nabla F_n(\mathbf{y}_{t,i}^n) - \nabla F_n(\mathbf{x}_t) + \nabla F_n(\mathbf{x}_t) - \nabla \tilde{f}(\mathbf{x}_t) + \nabla \tilde{f}(\mathbf{x}_t) \right) \right\|^2 \right]$$

$$\leq \mathbb{E}_t \left[ \left\| \gamma \left( \mathbf{g}_n(\mathbf{y}_{t,i}^n) - \nabla F_n(\mathbf{y}_{t,i}^n) \right) \right\|^2 \right] + \left( 1 + \frac{1}{2I - 1} \right) \mathbb{E}_t \left[ \left\| \mathbf{y}_{t,i}^n - \mathbf{x}_t \right\|^2 \right]$$

$$+ 6I \mathbb{E}_t \left[ \left\| \gamma \left( \nabla F_n(\mathbf{y}_{t,i}^n) - \nabla F_n(\mathbf{x}_t) \right) \right\|^2 \right] + 6I \mathbb{E}_t \left[ \left\| \gamma \left( \nabla F_n(\mathbf{x}_t) - \nabla \tilde{f}(\mathbf{x}_t) \right) \right\|^2 \right]$$

$$+ 6I \left\| \gamma \nabla \tilde{f}(\mathbf{x}_t) \right\|^2$$

$$\overset{(e)}{\leq} \gamma^2 \sigma^2 + \left( 1 + \frac{1}{2I - 1} \right) \mathbb{E}_t \left[ \left\| \mathbf{y}_{t,i}^n - \mathbf{x}_t \right\|^2 \right] + 6I\gamma^2 L^2 \mathbb{E}_t \left[ \left\| \mathbf{y}_{t,i}^n - \mathbf{x}_t \right\|^2 \right] + 6I\gamma^2 \tilde{\delta}^2$$

$$+ 6I\gamma^2 \left\| \nabla \tilde{f}(\mathbf{x}_t) \right\|^2$$

$$= \left( 1 + \frac{1}{2I - 1} + 6I\gamma^2 L^2 \right) \mathbb{E}_t \left[ \left\| \mathbf{y}_{t,i}^n - \mathbf{x}_t \right\|^2 \right] + \gamma^2 \sigma^2 + 6I\gamma^2 \tilde{\delta}^2 + 6I\gamma^2 \left\| \nabla \tilde{f}(\mathbf{x}_t) \right\|^2, \quad \text{(B.3.4)}$$

where $(a)$ follows from expanding the squared norm above and applying the law of total expectation on the second term, $(b)$ is because the second part of the inner product has no randomness when $\mathbf{y}_{t,i}^n$ and $\mathbf{x}_t$ are given, $(c)$ is because the inner product is zero due to the unbiasedness of stochastic gradient, $(d)$ follows from expanding the second term and applying the Peter-Paul inequality, $(e)$ uses gradient variance bound, Lipschitz gradient, and gradient divergence bound.

Because $\gamma \leq \frac{1}{\sqrt{30LI}}$, we have

$$\frac{1}{2I - 1} + 6I\gamma^2 L^2 \leq \frac{1}{2I - 1} + \frac{1}{5I} \leq \frac{2}{2I - 1} = \frac{1}{I - \frac{1}{2}}.$$

Continuing from (B.3.4), we have

$$\mathbb{E}_t \left[ \left\| \mathbf{y}_{t,i+1}^n - \mathbf{x}_t \right\|^2 \right] \leq \left( 1 + \frac{1}{I - \frac{1}{2}} \right) \mathbb{E}_t \left[ \left\| \mathbf{y}_{t,i}^n - \mathbf{x}_t \right\|^2 \right] + \gamma^2 \sigma^2 + 6I\gamma^2 \tilde{\delta}^2 + 6I\gamma^2 \left\| \nabla \tilde{f}(\mathbf{x}_t) \right\|^2$$

By unrolling the recursion, we obtain

$$\mathbb{E}_t \left[ \left\| \mathbf{y}_{t,i+1}^n - \mathbf{x}_t \right\|^2 \right]$$

$$\leq \sum_{i'=0}^{i} \left( 1 + \frac{1}{I - \frac{1}{2}} \right)^{i'} \left( \gamma^2 \sigma^2 + 6I\gamma^2 \tilde{\delta}^2 + 6I\gamma^2 \left\| \nabla \tilde{f}(\mathbf{x}_t) \right\|^2 \right)$$

$$\leq \sum_{i'=0}^{I-1} \left( 1 + \frac{1}{I - \frac{1}{2}} \right)^{i'} \left( \gamma^2 \sigma^2 + 6I\gamma^2 \tilde{\delta}^2 + 6I\gamma^2 \left\| \nabla \tilde{f}(\mathbf{x}_t) \right\|^2 \right)$$

$$= \left[ \left( 1 + \frac{1}{I - \frac{1}{2}} \right)^I - 1 \right] \left( I - \frac{1}{2} \right) \cdot \left( \gamma^2 \sigma^2 + 6I\gamma^2 \tilde{\delta}^2 + 6I\gamma^2 \left\| \nabla \tilde{f}(\mathbf{x}_t) \right\|^2 \right)$$

$$= \left[ \left(1 + \frac{1}{I - \frac{1}{2}}\right)^{I - \frac{1}{2}} \left(1 + \frac{1}{I - \frac{1}{2}}\right)^{\frac{1}{2}} - 1 \right] \left(I - \frac{1}{2}\right) \cdot \left(\gamma^2 \sigma^2 + 6I\gamma^2 \tilde{\delta}^2 + 6I\gamma^2 \left\| \nabla \tilde{f}(\mathbf{x}_t) \right\|^2\right)$$

$$\overset{(a)}{\leq} \left[\sqrt{3}e - 1\right] \left(I - \frac{1}{2}\right) \cdot \left(\gamma^2 \sigma^2 + 6I\gamma^2 \tilde{\delta}^2 + 6I\gamma^2 \left\| \nabla \tilde{f}(\mathbf{x}_t) \right\|^2\right)$$

$$\leq 5I\gamma^2 \left(\sigma^2 + 6I\tilde{\delta}^2\right) + 30I^2\gamma^2 \left\| \nabla \tilde{f}(\mathbf{x}_t) \right\|^2$$

where $(a)$ uses $\left(1 + \frac{1}{z}\right)^z \leq e$ for any $z > 0$ and $1 + \frac{1}{I - \frac{1}{2}} \leq 3$. $\qquad\square$

**Lemma B.3.3** (General descent lemma). *When* $\gamma \leq \frac{1}{4\sqrt{15}LI}$ *and* $\gamma\eta \leq \frac{1}{4LI}$, *we have*

$$\mathbb{E}_t \left[\tilde{f}(\mathbf{x}_{t+1})\right]$$

$$\leq \tilde{f}(\mathbf{x}_t) + \frac{3\gamma\eta I N}{2} \left(15L^2 I \gamma^2 \sigma^2 + \frac{27\tilde{\delta}^2}{8}\right) \sum_{n=1}^{N} \left(\frac{p_n \omega_t^n}{N} - \varphi_n\right)^2$$

$$+ \frac{5\gamma^3 \eta L^2 I^2}{2}(\sigma^2 + 6I\tilde{\delta}^2) + \frac{\gamma^2 \eta^2 LI}{N^2} \left(\frac{17\sigma^2}{16} + \frac{27I\tilde{\delta}^2}{8}\right) \sum_{n=1}^{N} p_n (\omega_t^n)^2$$

$$+ \gamma\eta I \left[\frac{81N}{16} \sum_{n=1}^{N} \left(\frac{p_n \omega_t^n}{N} - \varphi_n\right)^2 + 15L^2 I^2 \gamma^2 + \frac{27\gamma\eta LI}{8N^2} \sum_{n=1}^{N} p_n (\omega_t^n)^2 - \frac{1}{4}\right] \cdot \left\| \nabla \tilde{f}(\mathbf{x}_t) \right\|^2.$$

$$\text{(B.3.5)}$$

*Proof.* Due to Assumption 1 ($L$-smoothness), we have

$$\mathbb{E}_t \left[\tilde{f}(\mathbf{x}_{t+1})\right] \leq \tilde{f}(\mathbf{x}_t) - \gamma\eta \mathbb{E}_t \left[\left\langle \nabla \tilde{f}(\mathbf{x}_t), \frac{1}{N} \sum_{n=1}^{N} \mathbb{I}_t^n \omega_t^n \sum_{i=0}^{I-1} \mathbf{g}_n(\mathbf{y}_{t,i}^n) \right\rangle\right]$$

$$+ \frac{\gamma^2 \eta^2 L}{2} \mathbb{E}_t \left[\left\| \frac{1}{N} \sum_{n=1}^{N} \mathbb{I}_t^n \omega_t^n \sum_{i=0}^{I-1} \mathbf{g}_n(\mathbf{y}_{t,i}^n) \right\|^2\right]$$

$$= \tilde{f}(\mathbf{x}_t) - \gamma\eta \mathbb{E}_t \left[\left\langle \nabla \tilde{f}(\mathbf{x}_t), \frac{1}{N} \sum_{n=1}^{N} p_n \omega_t^n \sum_{i=0}^{I-1} \nabla F_n(\mathbf{y}_{t,i}^n) \right\rangle\right]$$

$$+ \frac{\gamma^2 \eta^2 L}{2} \mathbb{E}_t \left[\left\| \frac{1}{N} \sum_{n=1}^{N} \mathbb{I}_t^n \omega_t^n \sum_{i=0}^{I-1} \mathbf{g}_n(\mathbf{y}_{t,i}^n) \right\|^2\right], \qquad \text{(B.3.6)}$$

where the last equality is due to $\mathbb{E}_t \left[\mathbb{I}_t^n\right] = p_n$ and the unbiasedness of the stochastic gradient giving $\mathbb{E}_t \left[\mathbb{E}\left[\mathbf{g}_n(\mathbf{y}_{t,i}^n)\middle| \mathbf{y}_{t,i}^n\right]\right] = \mathbb{E}_t \left[\nabla F_n(\mathbf{y}_{t,i}^n)\right]$ (for simplicity, we will not write out this total expectation in subsequent steps of this proof).

Expanding the second term of (B.3.6), we have

$$- \gamma\eta \mathbb{E}_t \left[\left\langle \nabla \tilde{f}(\mathbf{x}_t), \frac{1}{N} \sum_{n=1}^{N} p_n \omega_t^n \sum_{i=0}^{I-1} \nabla F_n(\mathbf{y}_{t,i}^n) \right\rangle\right]$$

$$= -\gamma\eta \mathbb{E}_t \left[\left\langle \nabla \tilde{f}(\mathbf{x}_t), \frac{1}{N} \sum_{n=1}^{N} p_n \omega_t^n \sum_{i=0}^{I-1} \nabla F_n(\mathbf{y}_{t,i}^n) - \sum_{n=1}^{N} \varphi_n \sum_{i=0}^{I-1} \nabla F_n(\mathbf{y}_{t,i}^n) \right.\right.$$

$$\left.\left. + \sum_{n=1}^{N} \varphi_n \sum_{i=0}^{I-1} \nabla F_n(\mathbf{y}_{t,i}^n) - I\nabla \tilde{f}(\mathbf{x}_t) + I\nabla \tilde{f}(\mathbf{x}_t) \right\rangle\right]$$

$$= -\gamma\eta \mathbb{E}_t \left[\left\langle \nabla \tilde{f}(\mathbf{x}_t), \frac{1}{N} \sum_{n=1}^{N} p_n \omega_t^n \sum_{i=0}^{I-1} \nabla F_n(\mathbf{y}_{t,i}^n) - \sum_{n=1}^{N} \varphi_n \sum_{i=0}^{I-1} \nabla F_n(\mathbf{y}_{t,i}^n) \right\rangle\right]$$

$$- \gamma\eta\mathbb{E}_t\left[\left\langle \nabla\tilde{f}(\mathbf{x}_t), \sum_{n=1}^{N}\varphi_n\sum_{i=0}^{I-1}\nabla F_n(\mathbf{y}_{t,i}^n) - I\nabla\tilde{f}(\mathbf{x}_t)\right\rangle\right] - \gamma\eta I\left\|\nabla\tilde{f}(\mathbf{x}_t)\right\|^2$$

$$= -\frac{\gamma\eta}{I}\mathbb{E}_t\left[\left\langle I\nabla\tilde{f}(\mathbf{x}_t), \sum_{n=1}^{N}\left(\frac{p_n\omega_t^n}{N} - \varphi_n\right)\sum_{i=0}^{I-1}\nabla F_n(\mathbf{y}_{t,i}^n)\right\rangle\right]$$

$$-\frac{\gamma\eta}{I}\mathbb{E}_t\left[\left\langle I\nabla\tilde{f}(\mathbf{x}_t), \sum_{n=1}^{N}\varphi_n\sum_{i=0}^{I-1}(\nabla F_n(\mathbf{y}_{t,i}^n) - \nabla F_n(\mathbf{x}_t))\right\rangle\right] - \gamma\eta I\left\|\nabla\tilde{f}(\mathbf{x}_t)\right\|^2$$

$$\overset{(a)}{\leq} \frac{\gamma\eta I}{4}\left\|\nabla\tilde{f}(\mathbf{x}_t)\right\|^2 + \frac{\gamma\eta}{I}\mathbb{E}_t\left[\left\|\sum_{n=1}^{N}\left(\frac{p_n\omega_t^n}{N} - \varphi_n\right)\sum_{i=0}^{I-1}\nabla F_n(\mathbf{y}_{t,i}^n)\right\|^2\right]$$

$$+ \frac{\gamma\eta I}{2}\left\|\nabla\tilde{f}(\mathbf{x}_t)\right\|^2 + \frac{\gamma\eta}{2I}\mathbb{E}_t\left[\left\|\sum_{n=1}^{N}\varphi_n\sum_{i=0}^{I-1}(\nabla F_n(\mathbf{y}_{t,i}^n) - \nabla F_n(\mathbf{x}_t))\right\|^2\right]$$

$$-\frac{\gamma\eta}{2I}\mathbb{E}_t\left[\left\|\sum_{n=1}^{N}\varphi_n\sum_{i=0}^{I-1}\nabla F_n(\mathbf{y}_{t,i}^n)\right\|^2\right] - \gamma\eta I\left\|\nabla\tilde{f}(\mathbf{x}_t)\right\|^2$$

$$\leq \gamma\eta N\sum_{n=1}^{N}\left(\frac{p_n\omega_t^n}{N} - \varphi_n\right)^2\sum_{i=0}^{I-1}\mathbb{E}_t\left[\left\|\nabla F_n(\mathbf{y}_{t,i}^n)\right\|^2\right]$$

$$+ \frac{\gamma\eta}{2}\sum_{n=1}^{N}\varphi_n\sum_{i=0}^{I-1}\mathbb{E}_t\left[\left\|\nabla F_n(\mathbf{y}_{t,i}^n) - \nabla F_n(\mathbf{x}_t)\right\|^2\right]$$

$$-\frac{\gamma\eta}{2I}\mathbb{E}_t\left[\left\|\sum_{n=1}^{N}\varphi_n\sum_{i=0}^{I-1}\nabla F_n(\mathbf{y}_{t,i}^n)\right\|^2\right] - \frac{\gamma\eta I}{4}\left\|\nabla\tilde{f}(\mathbf{x}_t)\right\|^2$$

$$\leq \gamma\eta N\sum_{n=1}^{N}\left(\frac{p_n\omega_t^n}{N} - \varphi_n\right)^2\sum_{i=0}^{I-1}\mathbb{E}_t\left[\left\|\nabla F_n(\mathbf{y}_{t,i}^n)\right\|^2\right] + \frac{\gamma\eta L^2}{2}\sum_{n=1}^{N}\varphi_n\sum_{i=0}^{I-1}\mathbb{E}_t\left[\left\|\mathbf{y}_{t,i}^n - \mathbf{x}_t\right\|^2\right]$$

$$-\frac{\gamma\eta}{2I}\mathbb{E}_t\left[\left\|\sum_{n=1}^{N}\varphi_n\sum_{i=0}^{I-1}\nabla F_n(\mathbf{y}_{t,i}^n)\right\|^2\right] - \frac{\gamma\eta I}{4}\left\|\nabla\tilde{f}(\mathbf{x}_t)\right\|^2, \tag{B.3.7}$$

where we use $\langle\mathbf{a},\mathbf{b}\rangle = \frac{1}{2}(\|\mathbf{a}+\mathbf{b}\|^2 - \|\mathbf{a}\|^2 - \|\mathbf{b}\|^2)$ to expand the second term in $(a)$.

Expanding the third term of (B.3.6), we have

$$\frac{\gamma^2\eta^2 L}{2}\mathbb{E}_t\left[\left\|\frac{1}{N}\sum_{n=1}^{N}\mathbb{I}_t^n\omega_t^n\sum_{i=0}^{I-1}\mathbf{g}_n(\mathbf{y}_{t,i}^n)\right\|^2\right]$$

$$= \frac{\gamma^2\eta^2 L}{2}\mathbb{E}_t\left[\left\|\frac{1}{N}\sum_{n=1}^{N}\mathbb{I}_t^n\omega_t^n\sum_{i=0}^{I-1}\left(\mathbf{g}_n(\mathbf{y}_{t,i}^n) - \nabla F_n(\mathbf{y}_{t,i}^n)\right) + \frac{1}{N}\sum_{n=1}^{N}\mathbb{I}_t^n\omega_t^n\sum_{i=0}^{I-1}\nabla F_n(\mathbf{y}_{t,i}^n)\right\|^2\right]$$

$$\leq \gamma^2\eta^2 L\mathbb{E}_t\left[\left\|\frac{1}{N}\sum_{n=1}^{N}\mathbb{I}_t^n\omega_t^n\sum_{i=0}^{I-1}\left(\mathbf{g}_n(\mathbf{y}_{t,i}^n) - \nabla F_n(\mathbf{y}_{t,i}^n)\right)\right\|^2\right]$$

$$+ \gamma^2\eta^2 L\mathbb{E}_t\left[\left\|\frac{1}{N}\sum_{n=1}^{N}\mathbb{I}_t^n\omega_t^n\sum_{i=0}^{I-1}\nabla F_n(\mathbf{y}_{t,i}^n)\right\|^2\right]$$

$$\overset{(a)}{=} \frac{\gamma^2\eta^2 L}{N^2}\sum_{n=1}^{N}\mathbb{E}_t\left[\left\|\mathbb{I}_t^n\omega_t^n\sum_{i=0}^{I-1}\left(\mathbf{g}_n(\mathbf{y}_{t,i}^n) - \nabla F_n(\mathbf{y}_{t,i}^n)\right)\right\|^2\right]$$

$$+ \gamma^2 \eta^2 L \mathbb{E}_t \left[ \left\| \frac{1}{N} \sum_{n=1}^{N} (\mathbb{1}_t^n - p_n + p_n) \omega_t^n \sum_{i=0}^{I-1} \nabla F_n(\mathbf{y}_{t,i}^n) \right\|^2 \right]$$

$$\overset{(b)}{=} \frac{\gamma^2 \eta^2 L I \sigma^2}{N^2} \sum_{n=1}^{N} p_n (\omega_t^n)^2 + \gamma^2 \eta^2 L \mathbb{E}_t \left[ \left\| \frac{1}{N} \sum_{n=1}^{N} (\mathbb{1}_t^n - p_n) \omega_t^n \sum_{i=0}^{I-1} \nabla F_n(\mathbf{y}_{t,i}^n) \right\|^2 \right]$$

$$+ \gamma^2 \eta^2 L \mathbb{E}_t \left[ \left\| \frac{1}{N} \sum_{n=1}^{N} p_n \omega_t^n \sum_{i=0}^{I-1} \nabla F_n(\mathbf{y}_{t,i}^n) \right\|^2 \right]$$

$$\overset{(c)}{=} \frac{\gamma^2 \eta^2 L I \sigma^2}{N^2} \sum_{n=1}^{N} p_n (\omega_t^n)^2 + \frac{\gamma^2 \eta^2 L}{N^2} \sum_{n=1}^{N} \mathbb{E}_t \left[ \left\| (\mathbb{1}_t^n - p_n) \omega_t^n \sum_{i=0}^{I-1} \nabla F_n(\mathbf{y}_{t,i}^n) \right\|^2 \right]$$

$$+ \gamma^2 \eta^2 L \mathbb{E}_t \left[ \left\| \sum_{n=1}^{N} \left( \frac{p_n \omega_t^n}{N} - \varphi_n + \varphi_n \right) \sum_{i=0}^{I-1} \nabla F_n(\mathbf{y}_{t,i}^n) \right\|^2 \right]$$

$$\overset{(d)}{\leq} \frac{\gamma^2 \eta^2 L I \sigma^2}{N^2} \sum_{n=1}^{N} p_n (\omega_t^n)^2 + \frac{\gamma^2 \eta^2 L}{N^2} \sum_{n=1}^{N} p_n (1 - p_n) (\omega_t^n)^2 \mathbb{E}_t \left[ \left\| \sum_{i=0}^{I-1} \nabla F_n(\mathbf{y}_{t,i}^n) \right\|^2 \right]$$

$$+ 2 \gamma^2 \eta^2 L \mathbb{E}_t \left[ \left\| \sum_{n=1}^{N} \left( \frac{p_n \omega_t^n}{N} - \varphi_n \right) \sum_{i=0}^{I-1} \nabla F_n(\mathbf{y}_{t,i}^n) \right\|^2 \right] + 2 \gamma^2 \eta^2 L \mathbb{E}_t \left[ \left\| \sum_{n=1}^{N} \varphi_n \sum_{i=0}^{I-1} \nabla F_n(\mathbf{y}_{t,i}^n) \right\|^2 \right]$$

$$\leq \frac{\gamma^2 \eta^2 L I \sigma^2}{N^2} \sum_{n=1}^{N} p_n (\omega_t^n)^2 + \frac{\gamma^2 \eta^2 L I}{N^2} \sum_{n=1}^{N} p_n (\omega_t^n)^2 \sum_{i=0}^{I-1} \mathbb{E}_t \left[ \left\| \nabla F_n(\mathbf{y}_{t,i}^n) \right\|^2 \right]$$

$$+ 2 \gamma^2 \eta^2 L I N \sum_{n=1}^{N} \left( \frac{p_n \omega_t^n}{N} - \varphi_n \right)^2 \sum_{i=0}^{I-1} \mathbb{E}_t \left[ \left\| \nabla F_n(\mathbf{y}_{t,i}^n) \right\|^2 \right]$$

$$+ 2 \gamma^2 \eta^2 L \mathbb{E}_t \left[ \left\| \sum_{n=1}^{N} \varphi_n \sum_{i=0}^{I-1} \nabla F_n(\mathbf{y}_{t,i}^n) \right\|^2 \right], \tag{B.3.8}$$

where we note that $\mathbb{1}_t^n$ follows Bernoulli distribution with probability $p_n$, thus $\mathbb{E}\left[\mathbb{1}_t^n\right] = p_n$ and $\text{Var}\left[\mathbb{1}_t^n\right] = p_n(1 - p_n)$, yielding the relation in $(d)$. We also use the independence across different $n$ and $i$ for the stochastic gradients and the independence across $n$ for the client participation random variable $\mathbb{1}_t^n$, as well as the fact that $\mathbb{1}_t^n$ and the stochastic gradients are independent of each other, so the local updates (progression of $\mathbf{y}_{t,i}^n$) are independent of $\mathbb{1}_t^n$ according to the logically equivalent algorithm formulation in Algorithm B.2.1. The independence yields some inner product terms to be zero, giving the results in $(a)$, $(b)$, and $(c)$.

For $\mathbb{E}_t \left[ \left\| \nabla F_n(\mathbf{y}_{t,i}^n) \right\|^2 \right]$ that exists in both (B.3.7) and (B.3.8), we note that

$$\mathbb{E}_t \left[ \left\| \nabla F_n(\mathbf{y}_{t,i}^n) \right\|^2 \right]$$

$$= \mathbb{E}_t \left[ \left\| \nabla F_n(\mathbf{y}_{t,i}^n) - \nabla F_n(\mathbf{x}_t) + \nabla F_n(\mathbf{x}_t) - \nabla \tilde{f}(\mathbf{x}_t) + \nabla \tilde{f}(\mathbf{x}_t) \right\|^2 \right]$$

$$\leq 3 \mathbb{E}_t \left[ \left\| \nabla F_n(\mathbf{y}_{t,i}^n) - \nabla F_n(\mathbf{x}_t) \right\|^2 \right] + 3 \left\| \nabla F_n(\mathbf{x}_t) - \nabla \tilde{f}(\mathbf{x}_t) \right\|^2 + 3 \left\| \nabla \tilde{f}(\mathbf{x}_t) \right\|^2$$

$$\leq 3 L^2 \mathbb{E}_t \left[ \left\| \mathbf{y}_{t,i}^n - \mathbf{x}_t \right\|^2 \right] + 3 \tilde{\delta}^2 + 3 \left\| \nabla \tilde{f}(\mathbf{x}_t) \right\|^2$$

$$\leq 15 L^2 I \gamma^2 (\sigma^2 + 6 I \tilde{\delta}^2) + 3 \tilde{\delta}^2 + \left( 90 L^2 I^2 \gamma^2 + 3 \right) \left\| \nabla \tilde{f}(\mathbf{x}_t) \right\|^2. \tag{B.3.9}$$

Combining (B.3.7), (B.3.8), (B.3.9) gives

$$- \gamma\eta\mathbb{E}_t\left[\left\langle\nabla\tilde{f}(\mathbf{x}_t), \frac{1}{N}\sum_{n=1}^N p_n\omega_t^n\sum_{i=0}^{I-1}\nabla F_n(\mathbf{y}_{t,i}^n)\right\rangle\right] + \frac{\gamma^2\eta^2 L}{2}\mathbb{E}_t\left[\left\|\frac{1}{N}\sum_{n=1}^N \mathbb{I}_t^n\omega_t^n\sum_{i=0}^{I-1}\mathbf{g}_n(\mathbf{y}_{t,i}^n)\right\|^2\right]$$

$$\leq \gamma\eta N\sum_{n=1}^N\left(\frac{p_n\omega_t^n}{N} - \varphi_n\right)^2\sum_{i=0}^{I-1}\mathbb{E}_t\left[\|\nabla F_n(\mathbf{y}_{t,i}^n)\|^2\right] + \frac{\gamma\eta L^2}{2}\sum_{n=1}^N\varphi_n\sum_{i=0}^{I-1}\mathbb{E}_t\left[\|\mathbf{y}_{t,i}^n - \mathbf{x}_t\|^2\right]$$

$$- \frac{\gamma\eta}{2I}\mathbb{E}_t\left[\left\|\sum_{n=1}^N\varphi_n\sum_{i=0}^{I-1}\nabla F_n(\mathbf{y}_{t,i}^n)\right\|^2\right] - \frac{\gamma\eta I}{4}\left\|\nabla\tilde{f}(\mathbf{x}_t)\right\|^2$$

$$+ \frac{\gamma^2\eta^2 LI\sigma^2}{N^2}\sum_{n=1}^N p_n(\omega_t^n)^2 + \frac{\gamma^2\eta^2 LI}{N^2}\sum_{n=1}^N p_n(\omega_t^n)^2\sum_{i=0}^{I-1}\mathbb{E}_t\left[\|\nabla F_n(\mathbf{y}_{t,i}^n)\|^2\right]$$

$$+ 2\gamma^2\eta^2 LIN\sum_{n=1}^N\left(\frac{p_n\omega_t^n}{N} - \varphi_n\right)^2\sum_{i=0}^{I-1}\mathbb{E}_t\left[\|\nabla F_n(\mathbf{y}_{t,i}^n)\|^2\right]$$

$$+ 2\gamma^2\eta^2 L\mathbb{E}_t\left[\left\|\sum_{n=1}^N\varphi_n\sum_{i=0}^{I-1}\nabla F_n(\mathbf{y}_{t,i}^n)\right\|^2\right]$$

$$\leq \gamma\eta\left(1 + 2\gamma\eta LI\right)N\sum_{n=1}^N\left(\frac{p_n\omega_t^n}{N} - \varphi_n\right)^2\sum_{i=0}^{I-1}\mathbb{E}_t\left[\|\nabla F_n(\mathbf{y}_{t,i}^n)\|^2\right]$$

$$+ \frac{\gamma\eta L^2}{2}\sum_{n=1}^N\varphi_n\sum_{i=0}^{I-1}\mathbb{E}_t\left[\|\mathbf{y}_{t,i}^n - \mathbf{x}_t\|^2\right]$$

$$+ \frac{\gamma^2\eta^2 LI\sigma^2}{N^2}\sum_{n=1}^N p_n(\omega_t^n)^2 + \frac{\gamma^2\eta^2 LI}{N^2}\sum_{n=1}^N p_n(\omega_t^n)^2\sum_{i=0}^{I-1}\mathbb{E}_t\left[\|\nabla F_n(\mathbf{y}_{t,i}^n)\|^2\right]$$

$$- \left(\frac{\gamma\eta}{2I} - 2\gamma^2\eta^2 L\right)\mathbb{E}_t\left[\left\|\sum_{n=1}^N\varphi_n\sum_{i=0}^{I-1}\nabla F_n(\mathbf{y}_{t,i}^n)\right\|^2\right] - \frac{\gamma\eta I}{4}\left\|\nabla\tilde{f}(\mathbf{x}_t)\right\|^2$$

$$\overset{(a)}{\leq} \gamma\eta\left(1 + 2\gamma\eta LI\right)N\sum_{n=1}^N\left(\frac{p_n\omega_t^n}{N} - \varphi_n\right)^2\sum_{i=0}^{I-1}\mathbb{E}_t\left[\|\nabla F_n(\mathbf{y}_{t,i}^n)\|^2\right]$$

$$+ \frac{\gamma\eta L^2}{2}\sum_{n=1}^N\varphi_n\sum_{i=0}^{I-1}\mathbb{E}_t\left[\|\mathbf{y}_{t,i}^n - \mathbf{x}_t\|^2\right]$$

$$+ \frac{\gamma^2\eta^2 LI\sigma^2}{N^2}\sum_{n=1}^N p_n(\omega_t^n)^2 + \frac{\gamma^2\eta^2 LI}{N^2}\sum_{n=1}^N p_n(\omega_t^n)^2\sum_{i=0}^{I-1}\mathbb{E}_t\left[\|\nabla F_n(\mathbf{y}_{t,i}^n)\|^2\right] - \frac{\gamma\eta I}{4}\left\|\nabla\tilde{f}(\mathbf{x}_t)\right\|^2$$

$$\overset{(b)}{\leq} \gamma\eta\left(1 + 2\gamma\eta LI\right)N\sum_{n=1}^N\left(\frac{p_n\omega_t^n}{N} - \varphi_n\right)^2\left[15L^2 I\gamma^2(\sigma^2 + 6I\tilde{\delta}^2) + 3\tilde{\delta}^2 + \left(90L^2 I^2\gamma^2 + 3\right)\left\|\nabla\tilde{f}(\mathbf{x}_t)\right\|^2\right]$$

$$+ \frac{\gamma\eta L^2 I}{2}\left[5I\gamma^2(\sigma^2 + 6I\tilde{\delta}^2) + 30I^2\gamma^2\left\|\nabla\tilde{f}(\mathbf{x}_t)\right\|^2\right]$$

$$+ \frac{\gamma^2\eta^2 LI\sigma^2}{N^2}\sum_{n=1}^N p_n(\omega_t^n)^2$$

$$+ \frac{\gamma^2\eta^2 LI^2}{N^2}\sum_{n=1}^N p_n(\omega_t^n)^2\left[15L^2 I\gamma^2(\sigma^2 + 6I\tilde{\delta}^2) + 3\tilde{\delta}^2 + \left(90L^2 I^2\gamma^2 + 3\right)\left\|\nabla\tilde{f}(\mathbf{x}_t)\right\|^2\right]$$

$$- \frac{\gamma\eta I}{4}\left\|\nabla\tilde{f}(\mathbf{x}_t)\right\|^2$$

$$\overset{(c)}{\le} \frac{3\gamma\eta IN}{2} \sum_{n=1}^{N} \left(\frac{p_n\omega_t^n}{N} - \varphi_n\right)^2 \left[15L^2I\gamma^2\sigma^2 + \frac{27\tilde{\delta}^2}{8} + \frac{27}{8}\left\|\nabla\tilde{f}(\mathbf{x}_t)\right\|^2\right]$$

$$+ \frac{\gamma\eta L^2 I}{2}\left[5I\gamma^2(\sigma^2 + 6I\tilde{\delta}^2) + 30I^2\gamma^2\left\|\nabla\tilde{f}(\mathbf{x}_t)\right\|^2\right]$$

$$+ \frac{\gamma^2\eta^2 LI\sigma^2}{N^2}\sum_{n=1}^{N} p_n(\omega_t^n)^2 + \frac{\gamma^2\eta^2 LI^2}{N^2}\sum_{n=1}^{N} p_n(\omega_t^n)^2\left[\frac{\sigma^2}{16I} + \frac{27\tilde{\delta}^2}{8} + \frac{27}{8}\left\|\nabla\tilde{f}(\mathbf{x}_t)\right\|^2\right]$$

$$- \frac{\gamma\eta I}{4}\left\|\nabla\tilde{f}(\mathbf{x}_t)\right\|^2$$

$$\le \frac{3\gamma\eta IN}{2}\left(15L^2I\gamma^2\sigma^2 + \frac{27\tilde{\delta}^2}{8}\right)\sum_{n=1}^{N}\left(\frac{p_n\omega_t^n}{N} - \varphi_n\right)^2$$

$$+ \frac{5\gamma^3\eta L^2 I^2}{2}(\sigma^2 + 6I\tilde{\delta}^2) + \frac{\gamma^2\eta^2 LI}{N^2}\left(\frac{17\sigma^2}{16} + \frac{27I\tilde{\delta}^2}{8}\right)\sum_{n=1}^{N} p_n(\omega_t^n)^2$$

$$+ \gamma\eta I\left[\frac{81N}{16}\sum_{n=1}^{N}\left(\frac{p_n\omega_t^n}{N} - \varphi_n\right)^2 + 15L^2I^2\gamma^2 + \frac{27\gamma\eta LI}{8N^2}\sum_{n=1}^{N} p_n(\omega_t^n)^2 - \frac{1}{4}\right]\cdot\left\|\nabla\tilde{f}(\mathbf{x}_t)\right\|^2,$$

$$\text{(B.3.10)}$$

where $(a)$ uses $\gamma\eta \le \frac{1}{4LI}$, $(b)$ uses Lemma B.3.2 and (B.3.9), and $(c)$ uses $\gamma\eta \le \frac{1}{4LI}$ and $\gamma \le \frac{1}{4\sqrt{15}LI}$. The final result is obtained by plugging (B.3.10) into (B.3.6). □

## B.4 FORMAL VERSION AND PROOF OF THEOREM 1

We first state the formal version of Theorem 1 as follows.

**Theorem B.4.1** (Objective minimized at convergence, formal). *Define an alternative objective function as*

$$h(\mathbf{x}) := \frac{1}{P}\sum_{n=1}^{N}\omega_n p_n F_n(\mathbf{x}), \tag{B.4.1}$$

*where $P := \sum_{n=1}^{N}\omega_n p_n$. Under Assumptions 1–4, when $\omega_t^n = \omega_n$ with some $\omega_n > 0$ for all $t$ and $n$, choosing $\gamma \le \frac{c}{\sqrt{T}}$ and $\eta > 0$ so that $\gamma\eta = \frac{c'}{\sqrt{T}}$ for some constants $c, c' > 0$, there exists a sufficiently large $T$ so that the result $\{\mathbf{x}_t\}$ obtained from Algorithm 1 satisfies $\frac{1}{T}\sum_{t=0}^{T-1}\mathbb{E}\left[\|\nabla h(\mathbf{x}_t)\|^2\right] \le \epsilon^2$ for any $\epsilon > 0$.*

*Proof.* According to Algorithm B.2.1, the result remains the same when we replace $\eta$ and $\omega_n$ (thus $\omega_t^n$) with $\tilde{\eta}$ and $\tilde{\alpha}_n$, respectively, while keeping the product $\tilde{\eta}\tilde{\alpha}_n = \eta\omega_n$. We choose $\tilde{\alpha}_n = \frac{\omega_n N}{P}$ and $\tilde{\eta} = \frac{\eta P}{N}$. Then, we choose $\varphi_n = \frac{\omega_n p_n}{P} = \frac{p_n\tilde{\alpha}_n}{N}$ in Lemma B.3.3. We can see that this choice satisfies $\sum_{n=1}^{N}\varphi_n = 1$, so Lemma B.3.3 holds after replacing $\eta$ and $\omega_t^n$ in the lemma with $\tilde{\eta}$ and $\tilde{\alpha}_n$, respectively, and $\tilde{f}(\mathbf{x})$ in Lemma B.3.3 is equal to $h(\mathbf{x})$ defined in Theorem B.4.1 with this choice of $\varphi_n$. Therefore,

$$\mathbb{E}_t\left[h(\mathbf{x}_{t+1})\right] \le h(\mathbf{x}_t) + \frac{5\gamma^3\tilde{\eta}L^2I^2}{2}(\sigma^2 + 6I\tilde{\delta}^2) + \frac{\gamma^2\tilde{\eta}^2 LI}{N^2}\left(\frac{17\sigma^2}{16} + \frac{27I\tilde{\delta}^2}{8}\right)\sum_{n=1}^{N} p_n\tilde{\alpha}_n^2$$

$$+ \gamma\tilde{\eta}I\left[15L^2I^2\gamma^2 + \frac{27\gamma\tilde{\eta}LI}{8N^2}\sum_{n=1}^{N} p_n\tilde{\alpha}_n^2 - \frac{1}{4}\right]\cdot\|\nabla h(\mathbf{x}_t)\|^2. \tag{B.4.2}$$

Because $\gamma \le \frac{c}{\sqrt{T}}$ and $\gamma\tilde{\eta} = \frac{c''}{\sqrt{T}}$, there exists a sufficiently large $T$ so that $15L^2I^2\gamma^2 + \frac{27\gamma\tilde{\eta}LI}{8N^2}\sum_{n=1}^{N} p_n\tilde{\alpha}_n^2 \le \frac{1}{8}$. In this case, after taking the total expectation of (B.4.2) and rearranging,

we have

$$
\mathbb{E}\left[\|\nabla h(\mathbf{x}_t)\|^2\right]
$$

$$
\leq \frac{8\left(\mathbb{E}\left[h(\mathbf{x}_t)\right] - \mathbb{E}\left[h(\mathbf{x}_{t+1})\right]\right)}{\gamma\tilde{\eta}I} + 20\gamma^2 L^2 I(\sigma^2 + 6I\tilde{\delta}^2) + \frac{8\gamma\tilde{\eta}L}{N^2}\left(\frac{17\sigma^2}{16} + \frac{27I\tilde{\delta}^2}{8}\right)\sum_{n=1}^{N} p_n\tilde{\alpha}_n^2.
$$

$$(B.4.3)$$

Then, summing up over $T$ rounds and dividing by $T$, we have

$$
\frac{1}{T}\sum_{t=0}^{T-1}\mathbb{E}\left[\|\nabla h(\mathbf{x}_t)\|^2\right] \leq \frac{8\mathcal{H}}{\gamma\tilde{\eta}IT} + 20\gamma^2 L^2 I(\sigma^2 + 6I\tilde{\delta}^2) + \frac{8\gamma\tilde{\eta}L}{N^2}\left(\frac{17\sigma^2}{16} + \frac{27I\tilde{\delta}^2}{8}\right)\sum_{n=1}^{N} p_n\tilde{\alpha}_n^2,
$$

$$(B.4.4)$$

where $\mathcal{H} := h(\mathbf{x}_0) - h^*$ with $h^* := \min_{\mathbf{x}} h(\mathbf{x})$ as the truly minimum value.

Since $\gamma \leq \frac{c}{\sqrt{T}}$ and $\gamma\tilde{\eta} = \frac{c''}{\sqrt{T}}$, we can see that the upper bound above converges to zero as $T \to \infty$. Thus, there exists a sufficiently large $T$ to achieve an upper bound of an arbitrarily positive value of $\epsilon^2$. $\qquad\square$

## B.5 PROOF OF THEOREM 2

We first present the following variant of the descent lemma for the original objective defined in (1).

**Lemma B.5.1** (Descent lemma for original objective). *Under the same conditions as in Lemma B.3.3,*

$$
\mathbb{E}_t\left[f(\mathbf{x}_{t+1})\right]
$$

$$
\leq f(\mathbf{x}_t) + \frac{3\gamma\eta I}{2N}\left(15L^2 I\gamma^2\sigma^2 + \frac{27\delta^2}{8}\right)\sum_{n=1}^{N}(p_n\omega_t^n - 1)^2 + \frac{5\gamma^3\eta L^2 I^2}{2}(\sigma^2 + 6I\delta^2)
$$

$$
+ \frac{\gamma^2\eta^2 LI}{N^2}\left(\frac{17\sigma^2}{16} + \frac{27I\delta^2}{8}\right)\sum_{n=1}^{N} p_n(\omega_t^n)^2
$$

$$
+ \gamma\eta I\left[\frac{81}{16N}\sum_{n=1}^{N}(p_n\omega_t^n - 1)^2 + 15L^2 I^2\gamma^2 + \frac{27\gamma\eta LI}{8N^2}\sum_{n=1}^{N} p_n(\omega_t^n)^2 - \frac{1}{4}\right] \cdot \|\nabla f(\mathbf{x}_t)\|^2.
$$

$$(B.5.1)$$

*Proof.* The result can be immediately obtained from Lemma B.3.3 by choosing $\varphi_n = \frac{1}{N}$ in (B.3.1) and noting that gradient divergence bound holds for $\delta$ with this choice of $\varphi_n$ according to Assumption 3. $\qquad\square$

*Proof of Theorem 2.* Consider the last term in Lemma B.5.1. Due to $\gamma \leq \frac{1}{4\sqrt{15}LI}$, and $\gamma\eta \leq \min\left\{\frac{1}{4LI}; \frac{N}{54LIQ}\right\}$ as specified in the theorem, we have $15L^2 I^2\gamma^2 \leq \frac{1}{16}$ and $\frac{27\gamma\eta LI}{8N^2}\sum_{n=1}^{N} p_n(\omega_t^n)^2 \leq \frac{1}{16}$.

*Case 1:* When assuming $\|\nabla f(\mathbf{x})\|^2 \leq G^2$, we have

$$
\gamma\eta I\left[\frac{81}{16N}\sum_{n=1}^{N}(p_n\omega_t^n - 1)^2 + 15L^2 I^2\gamma^2 + \frac{27\gamma\eta LI}{8N^2}\sum_{n=1}^{N} p_n(\omega_t^n)^2 - \frac{1}{4}\right] \cdot \|\nabla f(\mathbf{x}_t)\|^2
$$

$$
\leq \frac{81\gamma\eta IG^2}{16N}\sum_{n=1}^{N}(p_n\omega_t^n - 1)^2 - \frac{\gamma\eta I}{8}\|\nabla f(\mathbf{x}_t)\|^2.
$$

$$(B.5.2)$$

Plugging back into Lemma B.5.1, after taking total expectation and rearranging, we obtain

$$
\mathbb{E}\left[\|\nabla f(\mathbf{x}_t)\|^2\right]
$$

$$\leq \frac{8(\mathbb{E}\left[f(\mathbf{x}_t)\right] - \mathbb{E}\left[f(\mathbf{x}_{t+1})\right])}{\gamma\eta I} + \frac{12}{N}\left(15L^2 I\gamma^2\sigma^2 + \frac{27(\delta^2 + G^2)}{8}\right)\sum_{n=1}^{N}(p_n\omega_t^n - 1)^2$$

$$+ 20\gamma^2 L^2 I(\sigma^2 + 6I\delta^2) + \frac{\gamma\eta L}{N^2}\left(\frac{17\sigma^2}{2} + 27I\delta^2\right)\sum_{n=1}^{N}p_n(\omega_t^n)^2. \tag{B.5.3}$$

*Case 2:* When assuming $\frac{1}{N}\sum_{n=1}^{N}(p_n\omega_t^n - 1)^2 \leq \frac{1}{81}$, we have

$$\gamma\eta I\left[\frac{81}{16N}\sum_{n=1}^{N}(p_n\omega_t^n - 1)^2 + 15L^2 I^2\gamma^2 + \frac{27\gamma\eta LI}{8N^2}\sum_{n=1}^{N}p_n(\omega_t^n)^2 - \frac{1}{4}\right]\cdot\|\nabla f(\mathbf{x}_t)\|^2$$

$$\leq -\frac{\gamma\eta I}{16}\|\nabla f(\mathbf{x}_t)\|^2. \tag{B.5.4}$$

Plugging back into Lemma B.5.1, after taking total expectation and rearranging, we obtain

$$\mathbb{E}\left[\|\nabla f(\mathbf{x}_t)\|^2\right]$$

$$\leq \frac{16(\mathbb{E}\left[f(\mathbf{x}_t)\right] - \mathbb{E}\left[f(\mathbf{x}_{t+1})\right])}{\gamma\eta I} + \frac{24}{N}\left(15L^2 I\gamma^2\sigma^2 + \frac{27\delta^2}{8}\right)\sum_{n=1}^{N}(p_n\omega_t^n - 1)^2$$

$$+ 40\gamma^2 L^2 I(\sigma^2 + 6I\delta^2) + \frac{\gamma\eta L}{N^2}\left(17\sigma^2 + 54I\delta^2\right)\sum_{n=1}^{N}p_n(\omega_t^n)^2. \tag{B.5.5}$$

The final result is obtained by summing up either (B.5.3) or (B.5.5) over $T$ rounds and dividing by $T$, choosing $\Psi_G$ accordingly for each case, and absorbing the constants in $\mathcal{O}(\cdot)$ notation. $\qquad\square$

## B.6 PROOF OF THEOREM 3

We start by analyzing the statistical properties of the possibly cutoff participation interval $S_n$. Because in every round, each client $n$ participates according to a Bernoulli distribution with probability $p_n$, the random variable $S_n$ has the following probability distribution:

$$\Pr\{S_n = k\} = \begin{cases} p_n(1-p_n)^{k-1}, & \text{if } 1 \leq k < K \\ (1-p_n)^{k-1}, & \text{if } k = K \end{cases}, \tag{B.6.1}$$

which is a *"cutoff" geometric distribution* with a maximum value of $K$. We will refer to this probability distribution as $K$-cutoff geometric distribution. We can see that when $K \to \infty$, this distribution becomes the same as the geometric distribution, but we consider the general case with an arbitrary $K$ that is specified later. We also recall that the actual value of $p_n$ is unknown to the system, which is why we need to compute $\{\omega_t^n\}$ using the estimation procedure in Algorithm 2.

**Lemma B.6.1.** *Equation* (B.6.1) *defines a probability distribution, and the mean and variance of $S_n$ are*

$$\mathbb{E}\left[S_n\right] = \frac{1}{p_n} - \frac{(1-p_n)^K}{p_n}; \quad \text{Var}\left[S_n\right] = \frac{1-p_n}{p_n^2} - \frac{(2K-1)(1-p_n)^K}{p_n} - \frac{(1-p_n)^{2K}}{p_n^2}. \tag{B.6.2}$$

*Proof.* We first show that (B.6.1) defines a probability distribution. According to the definition in (B.6.1), we have $\sum_{k=1}^{K}\Pr\{S_n = k\} = 1$ for any $K$. We prove this by induction. Let $S_n$ and $S_n'$ denote the random variables following $K$-cutoff and $(K+1)$-cutoff geometric distributions, respectively. For $K = 2$, we have $\sum_{k=1}^{K}\Pr\{S_n = k\} = p_n + (1-p_n) = 1$. Therefore, we can assume that $\sum_{k=1}^{K}\Pr\{S_n = k\} = 1$ holds for a certain value of $K$. For $(K+1)$-cutoff distribution, we first note that according to (B.6.1),

$$\Pr\{S_n' = k\} = \begin{cases} \Pr\{S_n = k\}, & 1 \leq k < K \\ p_n\cdot\Pr\{S_n = K\}, & k = K \\ (1-p_n)\cdot\Pr\{S_n = K\}, & k = K+1 \end{cases}.$$

Therefore,

$$\sum_{k=1}^{K+1} \Pr\{S'_n = k\} = \sum_{k=1}^{K-1} \Pr\{S_n = k\} + p_n \cdot \Pr\{S_n = K\} + (1 - p_n) \cdot \Pr\{S_n = K\}$$

$$= \sum_{k=1}^{K} \Pr\{S_n = k\}$$

$$= 1$$

This shows that $\Pr\{S_n = k\}$ defined in (B.6.1) is a probability distribution.

In the following, we derive the mean and variance of $S_n$, where we use $\frac{dy}{dx}$ to denote the derivative of $y$ with respect to $x$.

We have

$$\mathbb{E}\left[S_n\right] = \sum_{k=1}^{K-1} k p_n (1 - p_n)^{k-1} + K(1 - p_n)^{K-1}$$

$$= p_n \left[ \frac{d}{dp_n} \left( -\sum_{k=1}^{K-1} (1 - p_n)^k \right) \right] + K(1 - p_n)^{K-1}$$

$$= p_n \left[ \frac{d}{dp_n} \left( \frac{(1 - p_n)\left((1 - p_n)^{K-1} - 1\right)}{p_n} \right) \right] + K(1 - p_n)^{K-1}$$

$$= p_n \left[ \frac{d}{dp_n} \left( \frac{(1 - p_n)^K - 1 + p_n}{p_n} \right) \right] + K(1 - p_n)^{K-1}$$

$$= p_n \left[ \frac{d}{dp_n} \left( \frac{(1 - p_n)^K}{p_n} - \frac{1}{p_n} + 1 \right) \right] + K(1 - p_n)^{K-1}$$

$$= p_n \left[ -\frac{K(1 - p_n)^{K-1} p_n + (1 - p_n)^K}{p_n^2} + \frac{1}{p_n^2} \right] + K(1 - p_n)^{K-1}$$

$$= \frac{1}{p_n} - \frac{(1 - p_n)^K}{p_n}, \tag{B.6.3}$$

which gives the expression for the expected value.

To compute the variance, we note that

$$\mathbb{E}\left[S_n(S_n - 1)\right]$$

$$= \sum_{k=1}^{K-1} k(k-1) p_n (1 - p_n)^{k-1} + K(K-1)(1 - p_n)^{K-1}$$

$$= p_n \left[ \frac{d}{dp_n} \left( -\sum_{k=1}^{K-1} (k-1)(1 - p_n)^k \right) \right] + K(K-1)(1 - p_n)^{K-1}$$

$$= p_n \left[ \frac{d}{dp_n} \left( -(1 - p_n)^2 \sum_{k=1}^{K-1} (k-1)(1 - p_n)^{k-2} \right) \right] + K(K-1)(1 - p_n)^{K-1}$$

$$= p_n \left[ \frac{d}{dp_n} \left( -(1 - p_n)^2 \frac{d}{dp_n} \left( -\sum_{k=1}^{K-1} (1 - p_n)^{k-1} \right) \right) \right] + K(K-1)(1 - p_n)^{K-1}$$

$$= p_n \left[ \frac{d}{dp_n} \left( (1 - p_n)^2 \frac{d}{dp_n} \left( \frac{1 - (1 - p_n)^{K-1}}{p_n} \right) \right) \right] + K(K-1)(1 - p_n)^{K-1}$$

$$= p_n \left[ \frac{d}{dp_n} \left( (1 - p_n)^2 \cdot \frac{(K-1)(1 - p_n)^{K-2} p_n - 1 + (1 - p_n)^{K-1}}{p_n^2} \right) \right] + K(K-1)(1 - p_n)^{K-1}$$

$$= p_n \left[ \frac{d}{dp_n} \left( \frac{(K-1)(1 - p_n)^K}{p_n} + \frac{(1 - p_n)^{K+1}}{p_n^2} - \frac{(1 - p_n)^2}{p_n^2} \right) \right] + K(K-1)(1 - p_n)^{K-1}$$

$$
\begin{aligned}
&= p_n\Bigg[ -\frac{K(K-1)(1-p_n)^{K-1}p_n + (K-1)(1-p_n)^K}{p_n^2} \\
&\qquad\quad -\frac{(K+1)(1-p_n)^K p_n^2 + 2(1-p_n)^{K+1}p_n}{p_n^4} \\
&\qquad\quad +\frac{2(1-p_n)p_n^2 + 2(1-p_n)^2 p_n}{p_n^4} \Bigg] + K(K-1)(1-p_n)^{K-1} \\
&= -\frac{(K-1)(1-p_n)^K}{p_n} - \frac{(K+1)(1-p_n)^K}{p_n} - \frac{2(1-p_n)^{K+1}}{p_n^2} + \frac{2(1-p_n)}{p_n} + \frac{2(1-p_n)^2}{p_n^2} \\
&= -\frac{2K(1-p_n)^K}{p_n} - \frac{2(1-p_n)(1-p_n)^K}{p_n^2} + \frac{2(1-p_n)}{p_n^2} \\
&= -\frac{2K(1-p_n)^K}{p_n} - \frac{2(1-p_n)^K}{p_n^2} + \frac{2(1-p_n)^K}{p_n} + \frac{2(1-p_n)}{p_n^2} \\
&= -\frac{2(K-1)(1-p_n)^K}{p_n} - \frac{2(1-p_n)^K}{p_n^2} + \frac{2(1-p_n)}{p_n^2}.
\end{aligned}
\tag{B.6.4}
$$

Thus,

$$
\begin{aligned}
\mathbb{E}\left[S_n^2\right] &= \mathbb{E}\left[S_n(S_n-1)\right] + \mathbb{E}\left[S_n\right] \\
&= -\frac{2(K-1)(1-p_n)^K}{p_n} - \frac{2(1-p_n)^K}{p_n^2} + \frac{2(1-p_n)}{p_n^2} + \frac{1}{p_n} - \frac{(1-p_n)^K}{p_n} \\
&= -\frac{(2K-1)(1-p_n)^K}{p_n} - \frac{2(1-p_n)^K}{p_n^2} + \frac{2-p_n}{p_n^2}.
\end{aligned}
\tag{B.6.5}
$$

Therefore,

$$
\begin{aligned}
\mathrm{Var}\left[S_n\right] &= \mathbb{E}\left[S_n^2\right] - \mathbb{E}\left[S_n\right]^2 \\
&= -\frac{(2K-1)(1-p_n)^K}{p_n} - \frac{2(1-p_n)^K}{p_n^2} + \frac{2-p_n}{p_n^2} - \left(\frac{1}{p_n} - \frac{(1-p_n)^K}{p_n}\right)^2 \\
&= -\frac{(2K-1)(1-p_n)^K}{p_n} - \frac{2(1-p_n)^K}{p_n^2} + \frac{2-p_n}{p_n^2} - \frac{1}{p_n^2} + \frac{2(1-p_n)^K}{p_n^2} - \frac{(1-p_n)^{2K}}{p_n^2} \\
&= -\frac{(2K-1)(1-p_n)^K}{p_n} + \frac{1-p_n}{p_n^2} - \frac{(1-p_n)^{2K}}{p_n^2},
\end{aligned}
\tag{B.6.6}
$$

which gives the final variance result. $\qquad\square$

Now, we are ready to obtain an upper bound of the weight error term.

*Proof of Theorem 3.*

*Case 1:* According to Algorithm 2, we have $\omega_t^n = 1$ in the initial rounds before the first participation has occurred. This includes at least one round ($t=0$) and at most $K$ rounds. In these initial rounds, we have

$$
\mathbb{E}\left[\left(p_n \omega_t^n - 1\right)^2\right] \leq 1.
\tag{B.6.7}
$$

*Case 2:* For all the other rounds, $\omega_t^n$ is estimated based on at least one sample of $S_n$. Therefore, using the mean and variance expressions from Lemma B.6.1, we have the following for these rounds:

$$
\begin{aligned}
\mathbb{E}\left[\left(p_n \omega_t^n - 1\right)^2\right] &= (p_n)^2 \mathbb{E}\left[\left(\omega_t^n - \left(\frac{1}{p_n} - \frac{(1-p_n)^K}{p_n}\right) - \frac{(1-p_n)^K}{p_n}\right)^2\right] \\
&\stackrel{(a)}{=} (p_n)^2 \mathbb{E}\left[\left(\omega_t^n - \left(\frac{1}{p_n} - \frac{(1-p_n)^K}{p_n}\right)\right)^2\right] + (1-p_n)^{2K}
\end{aligned}
$$

$$\overset{(b)}{\leq} (p_n)^2 \cdot \frac{\text{Var}\,[S_n]}{\max\left\{\lfloor \frac{t}{K}\rfloor, 1\right\}} + (1-p_n)^{2K}$$

$$\leq (p_n)^2 \cdot \frac{2K\text{Var}\,[S_n]}{t} + (1-p_n)^{2K}$$

$$\overset{(c)}{\leq} \frac{2K(1-p_n)}{t} + (1-p_n)^{2K}, \tag{B.6.8}$$

where $(a)$ is because the inner product term is zero since the mean of $\omega_t^n$ is equal to $\frac{1}{p_n} - \frac{(1-p_n)^K}{p_n}$; $(b)$ is due to the definition of variance, the fact that we consider the computation of $\omega_t^n$ to be based on at least one sample of $S_n$, and for any round $t$ there are at least $\lfloor \frac{t}{K}\rfloor$ samples of $S_n$ due to the cutoff interval of length $K$; $(c)$ uses the upper bound of $\text{Var}\,[S_n] \leq \frac{1-p_n}{p_n^2}$.

We note that the bound (B.6.7) in Case 1 always applies for $t=0$, because we always have $\omega_t^n = 1$ for $t=0$ according to Algorithm 2. For rounds $0 < t < K$, either the bound (B.6.7) in Case 1 or the bound (B.6.8) in Case 2 applies, thus $\mathbb{E}\left[(p_n\omega_t^n - 1)^2\right]$ is upper bounded by the sum of both bounds in these rounds. Then, for $t \geq K$, the bound (B.6.8) in Case 2 applies. According to this fact, summing up the bounds for each round and dividing by $T$ gives

$$\frac{1}{T}\sum_{t=0}^{T-1}\mathbb{E}\left[(p_n\omega_t^n - 1)^2\right] \leq \frac{1}{T}\left[K + (T-1)(1-p_n)^{2K} + 2K(1-p_n)\sum_{t=1}^{T-1}\frac{1}{t}\right]$$

$$\leq \frac{K + 2K(1-p_n)\left(\log T + 1\right)}{T} + (1-p_n)^{2K}$$

$$\leq \frac{3K + 2K\log T}{T} + (1-p_n)^{2K}, \tag{B.6.9}$$

where we use the relation that $\sum_{t=1}^{T-1}\frac{1}{t} \leq \log T + 1$ for $T \geq 2$, and the logarithm is based on $e$.

The final result is obtained by averaging (B.6.9) over all $n$. $\qquad\square$

## B.7 Proof of Corollary 4

We first prove the upper bound of the weight error term in the following lemma.

**Lemma B.7.1.** *Choosing* $K = \lceil\log_c T\rceil$, *where* $c := \left(\frac{1}{1-p}\right)^2$ *and* $p := \min_n p_n$. *Define* $R := \frac{1}{\log c}$. *When* $T \geq 2$, *the aggregation weights* $\{\omega_t^n\}$ *obtained from Algorithm 2 satisfies*

$$\frac{1}{NT}\sum_{t=0}^{T-1}\sum_{n=1}^{N}\mathbb{E}\left[(p_n\omega_t^n - 1)^2\right] \leq \mathcal{O}\left(\frac{R\log^2 T}{T}\right). \tag{B.7.1}$$

*Proof.* Let $K = \lceil\log_c T\rceil$, we have

$$\frac{1}{N}\sum_{n=1}^{N}(1-p_n)^{2K} \leq (1-p)^{2K} = (1-p)^{2\lceil\log_c T\rceil} \leq (1-p)^{2\log_c T} = \left(\frac{1}{\left(\frac{1}{1-p}\right)^2}\right)^{\log_c T}$$

$$= \frac{1}{c^{\log_c T}} = \frac{1}{T} \leq \frac{\log_2^2 T}{T} \tag{B.7.2}$$

This shows that by choosing $K = \lceil\log_c T\rceil$ where $T \geq 2$, the RHS in (7) of Theorem 3 is upper bounded by $O\left(\frac{\log^2 T}{T\log c} + \frac{\log^2 T}{T}\right) = O\left(\frac{R\log^2 T}{T}\right)$, which proves the result. $\qquad\square$

*Proof of Corollary 4.* We note that

$$\gamma = \min\left\{\frac{1}{LI\sqrt{T}}; \frac{1}{4\sqrt{15}LI}\right\} \leq \frac{1}{LI\sqrt{T}}$$

$$\gamma\eta = \min\left\{\sqrt{\frac{\mathcal{F}N}{Q\left(I\delta^2 + \sigma^2\right)LIT}}; \frac{1}{4LI}; \frac{N}{54LIQ}\right\} \leq \sqrt{\frac{\mathcal{F}N}{Q\left(I\delta^2 + \sigma^2\right)LIT}}$$

$$\frac{1}{\gamma\eta} = \max\left\{\sqrt{\frac{Q\left(I\delta^2 + \sigma^2\right)LIT}{\mathcal{F}N}}; 4LI; \frac{54LIQ}{N}\right\} \leq \sqrt{\frac{Q\left(I\delta^2 + \sigma^2\right)LIT}{\mathcal{F}N}} + 4LI + \frac{54LIQ}{N}$$

The result follows by plugging these upper bounds of $\gamma$, $\gamma\eta$, and $\frac{1}{\gamma\eta}$ and the result in Lemma B.7.1 into Theorem 2, where we note that $\sqrt{I\delta^2 + \sigma^2} \leq \sqrt{I}\delta + \sigma$ since $I\delta^2 + \sigma^2 \leq I\delta^2 + 2\sqrt{I}\delta\sigma + \sigma^2 = \left(\sqrt{I}\delta + \sigma\right)^2$. $\qquad\square$

# C    ADDITIONAL SETUP DETAILS OF EXPERIMENTS

## C.1    CODE

The code for reproducing our experiments is available via the following link:
`https://shiqiang.wang/code/fedau`

## C.2    DATASETS

The SVHN dataset has a citation requirement Netzer et al. (2011). Its license is for non-commercial use only. It includes $32 \times 32$ color images with real-world house numbers of 10 different digits, containing $73,257$ training data samples and $26,032$ test data samples.

The CIFAR-10 dataset only has a citation requirement Krizhevsky & Hinton (2009). It includes $32 \times 32$ color images of 10 different types of real-world objects, containing $50,000$ training data samples and $10,000$ test data samples.

The CIFAR-100 dataset only has a citation requirement Krizhevsky & Hinton (2009). It includes $32 \times 32$ color images of 100 different types of real-world objects, containing $50,000$ training data samples and $10,000$ test data samples.

The CINIC-10 dataset Darlow et al. (2018) has MIT license. It includes $32 \times 32$ color images of 10 different types of real-world objects, containing $90,000$ training data samples and $90,000$ test data samples.

We have cited all the references in the main paper and conformed to all the license terms.

We applied some basic data augmentation techniques to these datasets during the training stage. For SVHN, we applied random cropping. For CIFAR-10 and CINIC-10, we applied both random cropping and random horizontal flipping. For CIFAR-100, we applied a combination of random sharpness adjustment, color jitter, random posterization, random equalization, random cropping, and random horizontal flipping.

## C.3    MODELS

All the models include two convolutional layers with a kernel size of 3, filter size of 32, and ReLU activation, where each convolutional layer is followed by a max-pool layer. The model for the SVHN dataset has two fully connected layers, while the models for the CIFAR-10/100 and CINIC-10 datasets have three fully connected layers. All the fully connected layers use ReLU activation, except for the last layer that is connected to softmax output. For CIFAR-100 and CINIC-10 datasets, a dropout layer (with dropout probability $p = 0.2$) is applied before each fully connected layer. We use Kaiming initialization for the weights. See the code for further details on model definition (the model class files are located inside the "`model/`" subfolder).

## C.4    HYPERPARAMETERS

For each dataset and algorithm, we conducted a grid search on the learning rates $\gamma$ and $\eta$ separately. The grid for the local step size $\gamma$ is $\{10^{-2}, 10^{-1.75}, 10^{-1.5}, 10^{-1.25}, 10^{-1}, 10^{-0.75}, 10^{-0.5}\}$ and the grid for the global step size $\eta$ is $\{10^0, 10^{0.25}, 10^{0.5}, 10^{0.75}, 10^1, 10^{1.25}, 10^{1.5}\}$. To reduce the complexity of the search, we first search for the value of $\gamma$ with $\eta = 1$, and then search for $\eta$ while fixing $\gamma$ to the value found in the first search. We consider the training loss at 500 rounds for determining the best $\gamma$ and $\eta$. The hyperparameters found from this search and used in our experiments are shown in Table C.4.1.

*Learning Rate Decay for CIFAR-100 Dataset.* Only for the CIFAR-100 dataset, we decay the local learning rate $\gamma$ by half every $1,000$ rounds, starting from the $10,000$-th round.

Table C.4.1: Values of hyperparameters $\gamma$ and $\eta$, where we use $10^{-1.25} \approx 0.0562, 10^{-1.5} \approx 0.0316, 10^{0.25} \approx 1.78$

| Dataset | SVHN | | CIFAR-10 | | CIFAR-100 | | CINIC-10 | |
|---|---|---|---|---|---|---|---|---|
| Method / Hyperparameter | $\gamma$ | $\eta$ | $\gamma$ | $\eta$ | $\gamma$ | $\eta$ | $\gamma$ | $\eta$ |
| FedAU (ours, $K \to \infty$) | 0.1 | 1.0 | 0.1 | 1.0 | 0.0562 | 1.78 | 0.1 | 1.0 |
| FedAU (ours, $K = 50$) | 0.1 | 1.0 | 0.1 | 1.0 | 0.0562 | 1.78 | 0.1 | 1.0 |
| Average participating | 0.0562 | 1.78 | 0.0562 | 1.78 | 0.0316 | 1.78 | 0.0562 | 1.78 |
| Average all | 0.1 | 10.0 | 0.1 | 10.0 | 0.0562 | 10.0 | 0.1 | 10.0 |
| FedVarp ($250\times$ memory) | 0.1 | 1.0 | 0.0562 | 1.0 | 0.0316 | 1.78 | 0.0562 | 1.0 |
| MIFA ($250\times$ memory) | 0.1 | 1.0 | 0.0562 | 1.0 | 0.0316 | 1.78 | 0.0562 | 1.0 |
| Known participation statistics | 0.1 | 1.0 | 0.0562 | 1.0 | 0.0316 | 1.78 | 0.0562 | 1.0 |

## C.5 COMPUTATION RESOURCES

The experiments were split between a desktop machine with RTX 3070 GPU and an internal GPU cluster. In our experiments, the total number of rounds is $2,000$ for SVHN, $10,000$ for CIFAR-10 and CINIC-10, and $20,000$ for CIFAR-100. Each experiment with $10,000$ rounds took approximately $4$ hours to complete, for one random seed on RTX 3070 GPU. The time taken for experiments with other number of rounds scales accordingly. We ran experiments with 5 different random seeds for each dataset and algorithm. It was possible to run multiple experiments simultaneously on the same GPU while not exceeding the GPU memory.

## C.6 HETEROGENEOUS PARTICIPATION ACROSS CLIENTS

### C.6.1 GENERATING PARTICIPATION PATTERNS

In each experiment with a specific simulation seed, we take only one sample of this Dirichlet distribution with parameter $\alpha_p$, which gives a probability vector $\mathbf{q} \sim \mathrm{Dir}(\alpha_p)$ that has a dimension equal to the total number of classes in the dataset.[2] The participation probability $p_n$ for each client $n$ is obtained by computing an inner product between $\mathbf{q}$ and the class distribution vector of the data at client $n$, and then dividing by a normalization factor. The rationale behind this approach is that the elements in $\mathbf{q}$ indicate how different classes contribute to the participation probability. For example, if the first element of $\mathbf{q}$ is large, it means that clients with a lot of data samples in the first class will have a high participation probability, and vice versa. Since the participation probabilities $\{p_n\}$ generated using this approach are random variables, the normalization ensures a certain mean participation probability, i.e., $\mathbb{E}[p_n]$, of any client $n$, which is set to $0.1$ in our experiments. We further cap the minimum value of any $p_n$ to be $0.02$.

Among the three participation patterns in our experiments, i.e., Bernoulli, Markovian, and cyclic, we maintain the same *stationary* participation probabilities $\{p_n\}$ for the clients, so the difference is in the temporal distribution of when a client participates, which is summarized as follows.

- For Bernoulli participation, in every round $t$, each client $n$ decides whether or not to participate according to a Bernoulli distribution with probability $p_n$. This decision is independent across time, i.e., independent across different rounds.

- For Markovian participation, each client participates according a two-state Markov chain, where the motivation is similar to cyclic participation (see next item below) but includes more randomness. We set the maximum transition probability of a client transitioning from not participating to participating to $0.05$. The initial state of the Markov chain is determined by a random sampling according to the stationary probability $p_n$, and the transition probabilities are determined in a way so that the same stationary probability is maintained across all the subsequent rounds.

- For cyclic participation, each client participates cyclically, i.e., it participates for a certain number of rounds and does not participate in the other rounds of a cycle. This setup has

---

[2]We use $\mathrm{Dir}(\alpha_p)$ to denote $\mathrm{Dir}(\boldsymbol{\alpha}_p)$ with all the elements in the vector $\boldsymbol{\alpha}_p$ equal to $\alpha_p$.

been used in existing works to simulate periodic behavior of client devices being charged (e.g., at night) (Cho et al., 2023; Ding et al., 2020; Eichner et al., 2019; Wang & Ji, 2022). We set each cycle to be 100 rounds. We apply a random initial offset to the cycle for each client, to simulate a stationary random process for each client's participation pattern.

Figure C.6.1 shows examples of these three types of participation patterns.

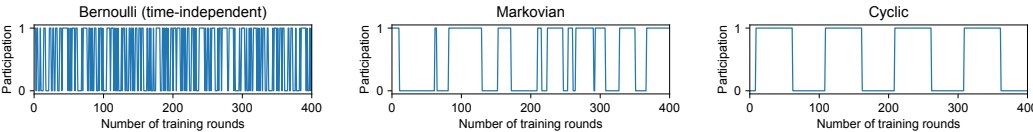

Figure C.6.1: Illustration of different participation patterns, in the first 400 rounds of a single client with 52.7% mean participation rate.

### C.6.2 ILLUSTRATION OF DATA AND PARTICIPATION HETEROGENEITY

As described in Section 5 and Appendix C.6.1, we generate the data and participation heterogeneity with two separate Dirichlet distributions with parameters $\alpha_d$ and $\alpha_p$, respectively. In the following, we illustrate the result of this generation for a specific random instance. In Figure C.6.2, the class-wise data distribution of each client is drawn from $\mathrm{Dir}(\alpha_d)$. For computing the participation probability, we draw a vector $\mathbf{q}$ from $\mathrm{Dir}(\alpha_p)$, which gave the following result in our random trial:

$$\mathbf{q} = [0.02, 0.05, 0.12, 0.00, 0.00, 0.78, 0.00, 0.00, 0.02, 0.00].$$

Then, the participation probability is set as the inner product of $\mathbf{q}$ and the class distribution of each client's data, divided by a normalization factor. For the above $\mathbf{q}$, the 6-th element has the highest value, which means that clients with a larger proportion of data in the 6-th class (label) will have a higher participation probability. This is confirmed by comparing the class distributions and the participation probabilities in Figure C.6.2.

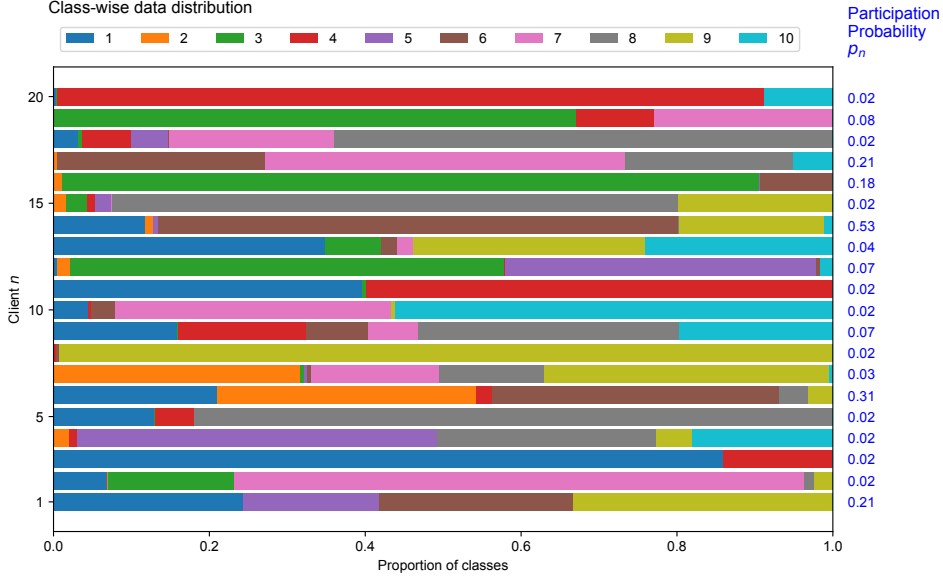

Figure C.6.2: Illustration of data and participation heterogeneity, for an example with 20 clients.

In this procedure, $\mathbf{q}$ is kept the same for all the clients, to simulate a consistent correlation between participation probability and class distribution across all the clients. However, the value of $\mathbf{q}$ changes with the random seed, which means that we have different $\mathbf{q}$ for different experiments. We ran

experiments with 5 different random seeds for each setting, which allows us to observe the general behavior.

More precisely, let $\boldsymbol{\kappa}_n \sim \mathrm{Dir}(\alpha_d)$ denote the class distribution of client $n$'s data. The participation probability $p_n$ of each client $n$ is computed as

$$p_n = \frac{1}{\lambda} \langle \boldsymbol{\kappa}_n, \mathbf{q} \rangle, \tag{C.6.1}$$

where $\lambda$ is the normalization factor to ensure that $\mathbb{E}\left[p_n\right]$ is equal to some target $\mu$, because $p_n$ is a random quantity when using this randomized generation procedure. In our experiments, we set $\mu = 0.1$. Let $C$ denote the total number of classes (labels). From the mean of Dirichlet distribution and the fact that $\boldsymbol{\kappa}_n$ and $\mathbf{q}$ are independent, we know that $\mathbb{E}\left[\langle \boldsymbol{\kappa}_n, \mathbf{q} \rangle\right] = \langle \mathbb{E}\left[\boldsymbol{\kappa}_n\right], \mathbb{E}\left[\mathbf{q}\right] \rangle = \frac{1}{C}$. Therefore, to ensure that $\mathbb{E}\left[p_n\right] = \mu$, according to (C.6.1), the normalization factor is chosen as $\lambda = \frac{1}{C\mu}$.

We emphasize again that this procedure is only used for simulating an experimental setup with both data and participation heterogeneity. Our FedAU algorithm still *does not* know the actual values of $\{p_n\}$.

# D  ADDITIONAL RESULTS FROM EXPERIMENTS

## D.1  RESULTS WITH DIFFERENT PARTICIPATION HETEROGENEITY

We present the results with different participation heterogeneity (characterized by the Dirichlet parameter $\alpha_p$) on the CIFAR-10 dataset in Table D.1.1, for the case of Bernoulli participation. The main observations remain consistent with those in Section 5. We also see that the difference between different methods becomes larger when the heterogeneity is higher (i.e., smaller $\alpha_p$), which aligns with intuition. For all degrees of heterogeneity, our FedAU algorithm performs the best among the algorithms that work under the same setting, i.e., the top part of Table D.1.1.

Table D.1.1: Accuracy results (in %) on training and test data of CIFAR-10, with different participation heterogeneity (Bernoulli participation)

| Participation heterogeneity | $\alpha_p = 0.01$ | | $\alpha_p = 0.05$ | | $\alpha_p = 0.1$ | | $\alpha_p = 0.5$ | | $\alpha_p = 1.0$ | |
|---|---|---|---|---|---|---|---|---|---|---|
| **Method / Metric** | Train | Test | Train | Test | Train | Test | Train | Test | Train | Test |
| FedAU (ours, $K \to \infty$) | 83.7±0.8 | 76.2±0.7 | 84.3±0.8 | 76.6±0.5 | 85.4±0.4 | 77.1±0.4 | 87.3±0.5 | **77.8**±0.2 | 88.1±0.7 | **78.1**±0.2 |
| FedAU (ours, $K = 50$) | **84.7**±0.6 | **76.9**±0.6 | **85.1**±0.5 | **77.1**±0.3 | **86.0**±0.5 | **77.3**±0.3 | **87.6**±0.4 | 77.8±0.4 | **88.2**±0.7 | 78.0±0.2 |
| Average participating | 80.6±1.2 | 72.3±1.7 | 81.5±1.1 | 72.6±1.4 | 83.5±0.9 | 74.1±0.8 | 85.9±0.7 | 75.7±0.9 | 87.0±1.0 | 76.8±0.6 |
| Average all | 76.9±2.7 | 69.5±2.7 | 78.5±1.7 | 70.6±1.8 | 81.0±0.9 | 72.7±0.9 | 83.6±1.4 | 74.6±0.8 | 84.9±1.2 | 75.9±1.0 |
| FedVarp (250× memory) | 82.5±0.8 | 77.3±0.3 | 83.0±0.4 | 77.5±0.4 | 84.2±0.3 | 77.9±0.2 | 85.4±0.5 | 78.1±0.2 | 86.4±0.7 | 78.5±0.3 |
| MIFA (250× memory) | 82.1±0.8 | 77.0±0.7 | 82.6±0.3 | 77.3±0.4 | 83.5±0.6 | 77.5±0.3 | 84.9±0.5 | 77.9±0.3 | 85.4±0.4 | 78.0±0.4 |
| Known participation statistics | 83.1±0.8 | 76.3±0.6 | 83.6±0.6 | 76.7±0.5 | 84.3±0.5 | 77.0±0.5 | 86.1±0.6 | 77.7±0.4 | 86.8±0.9 | 77.9±0.7 |

*Note to the table.* The same note in Table 1 also applies to this table.

## D.2  LOSS AND ACCURACY PLOTS

For Bernoulli participation, we plot the loss and accuracy results in different rounds for the four datasets, as shown in Figures D.2.1–D.2.4. In these plots, the curves show the mean values and the shaded areas show the standard deviation. We applied moving average with a window size equal to 3% of the total number of rounds, and the mean and standard deviation are computed across samples from all experiments (with 5 different random seeds) within each moving average window.

The main conclusions from Figures D.2.1–D.2.4 are similar to what we have seen from the final-round results shown in Table 1 in the main paper. We can see that our FedAU algorithm performs the best in the vast majority of cases and across most rounds. Only for the CIFAR-10 dataset, FedAU gives a slightly worse test accuracy compared to FedVarp and MIFA, which aligns with the results in Table 1. However, FedAU still gives the highest training accuracy on CIFAR-10. This implies that FedVarp/MIFA gives a slightly better generalization on the CIFAR-10 dataset, where the reasons are worth further investigation. We emphasize again that FedVarp and MIFA both require a substantial amount of additional memory than FedAU, thus they do not work under the same system assumptions as FedAU. For the CIFAR-100 dataset, there is a jump around the 10,000-th round due to the learning rate decay schedule, as mentioned in Section C.4.

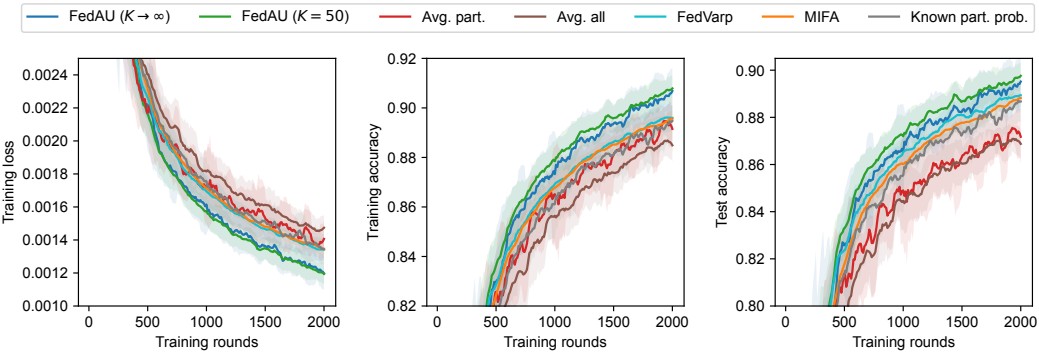

Figure D.2.1: Results on SVHN dataset (Bernoulli participation).

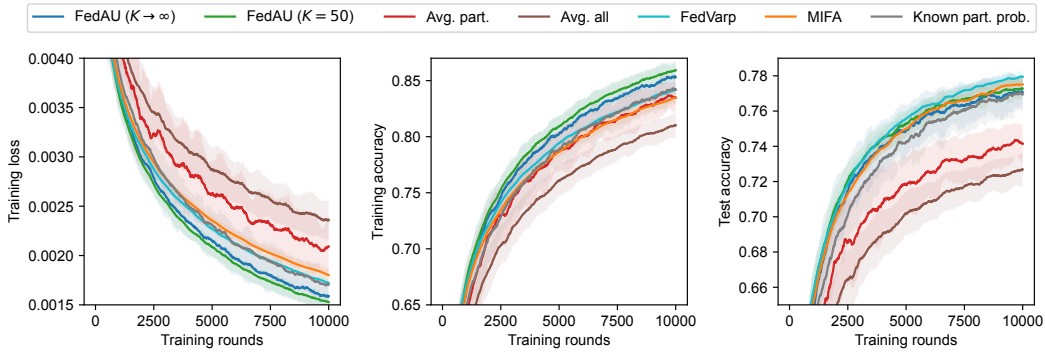

Figure D.2.2: Results on CIFAR-10 dataset (Bernoulli participation).

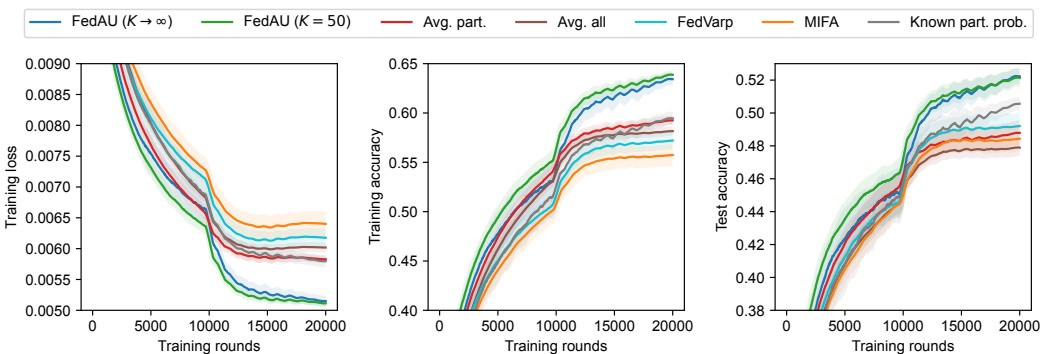

Figure D.2.3: Results on CIFAR-100 dataset (Bernoulli participation).

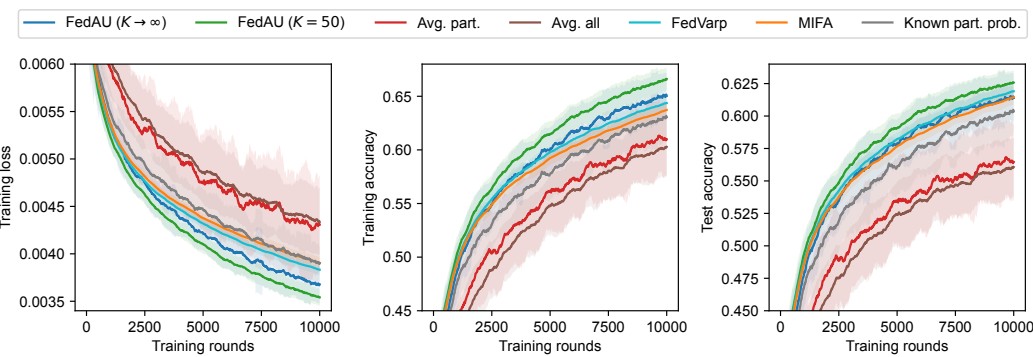

Figure D.2.4: Results on CINIC-10 dataset (Bernoulli participation).

## D.3 CLIENT-WISE DISTRIBUTIONS OF LOSS AND ACCURACY

We plot the loss and accuracy value distributions among all the clients in Figure D.3.1, where we consider Bernoulli participation and compare with baselines that *do not* require extra resources or information.

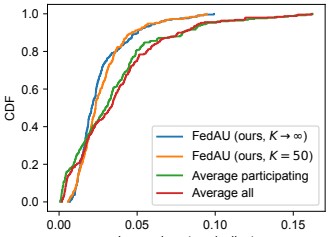 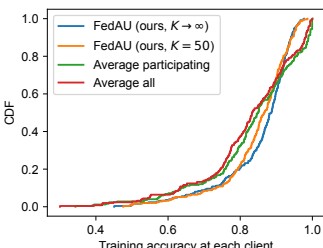 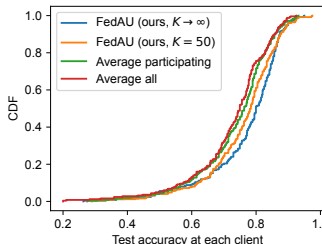

Figure D.3.1: Client-wise distributions of loss value, training accuracy, and test accuracy, on CIFAR-10 dataset with Bernoulli participation. The distribution is expressed as empirical cumulative distribution function (CDF).

We can see that compared to the average-participating and average-all baselines that use the same amount of memory as FedAU, the spread in the loss and accuracy with FedAU is smaller. This is also seen in the standard deviation of all the clients' loss and accuracy values in Table D.3.1, where we only include the standard deviation values because the mean values are the same as those in Table 1.

Table D.3.1: Client-wise statistics of loss and accuracy (CIFAR-10 dataset with Bernoulli participation)

| Method | Client-wise std. dev. of loss | Client-wise std. dev. of training accuracy | Client-wise std. dev. of test accuracy |
|---|---|---|---|
| FedAU (ours, $K \to \infty$) | 0.017 | 9.9% | 11.7% |
| FedAU (ours, $K = 50$) | **0.016** | **9.5%** | **11.2%** |
| Average participating | 0.031 | 13.3% | 11.6% |
| Average all | 0.030 | 13.3% | 12.2% |

This shows that FedAU (especially with $K = 50$) reduces the bias among clients compared to the two baselines, which aligns with our motivation mentioned in Section 1 about reducing discrimination.

## D.4 AGGREGATION WEIGHTS

As shown in Figure D.4.1, with Bernoulli participation, the computed weights can be quite different from $1/p_n$, especially when the participation probability is low (in Subfigures D.4.1c–D.4.1e). In contrast, we see in Figure D.4.2 that with cyclic participation the weights computed by FedAU and the known participation statistics baseline are more similar. This aligns with the fact that the accuracies in the case of cyclic participation are also more similar compared to the case of Bernoulli participation, as seen in Table 1.

Note that we use $K \to \infty$ for FedAU in both Figure D.4.1 and Figure D.4.2.

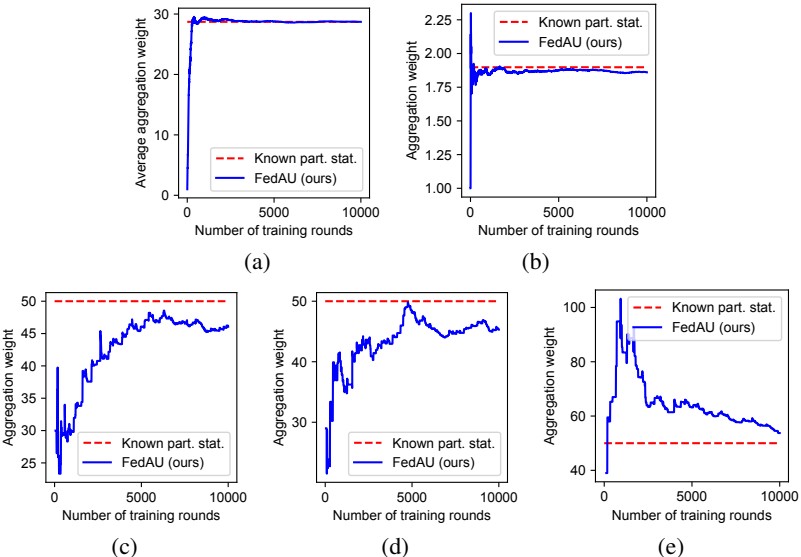

Figure D.4.1: Aggregation weights with Bernoulli participation, (a) average aggregation weights over all clients, (b) aggregation weights of a single client with a high mean participation rate of $52.7\%$, (c)-(e) aggregation weights of three individual clients with a low mean participation rate of $2\%$.

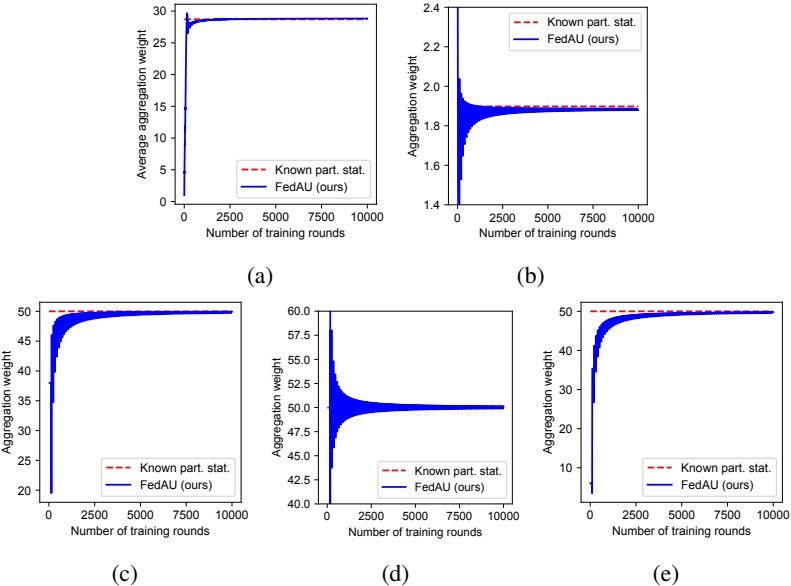

Figure D.4.2: Aggregation weights with cyclic participation, (a) average aggregation weights over all clients, (b) aggregation weights of a single client with a high mean participation rate of $52.7\%$, (c)-(e) aggregation weights of three individual clients with a low mean participation rate of $2\%$.

## D.5 CHOICE OF DIFFERENT $K$

We study the effect of the cutoff interval length $K$ by considering the performance of FedAU under different minimum participation probabilities. The distributions of participation probabilities $\{p_n\}$ for all the clients with different lower bounds are shown in Figure D.5.1, where we can see that a smaller lower bound value corresponds to having more clients with very small participation probabilities. The full set of plots complementing Figure 1 is shown in Figure D.5.2.

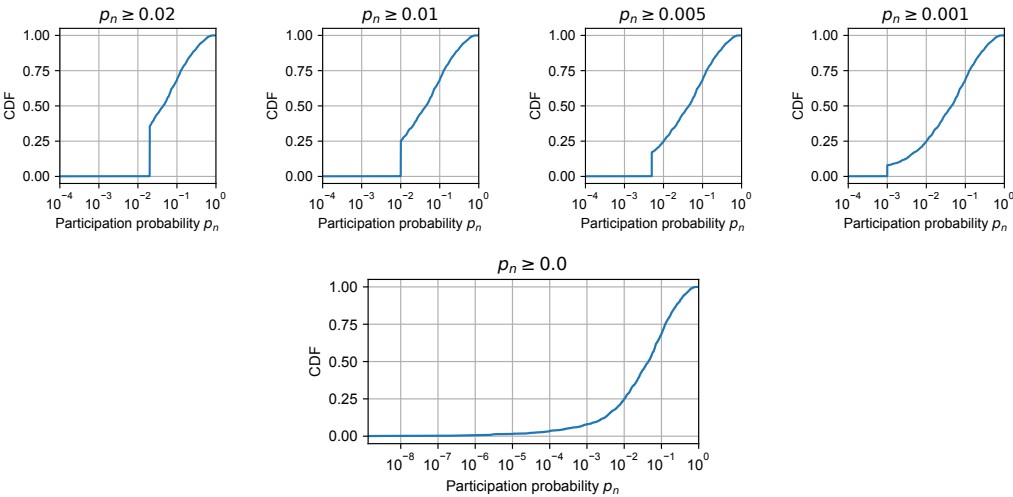

Figure D.5.1: Distributions of participation probabilities $\{p_n\}$ at clients, with different lower bound values of these probabilities. The distribution is expressed as empirical cumulative distribution function (CDF) with logarithmic scale on the $x$-axis.

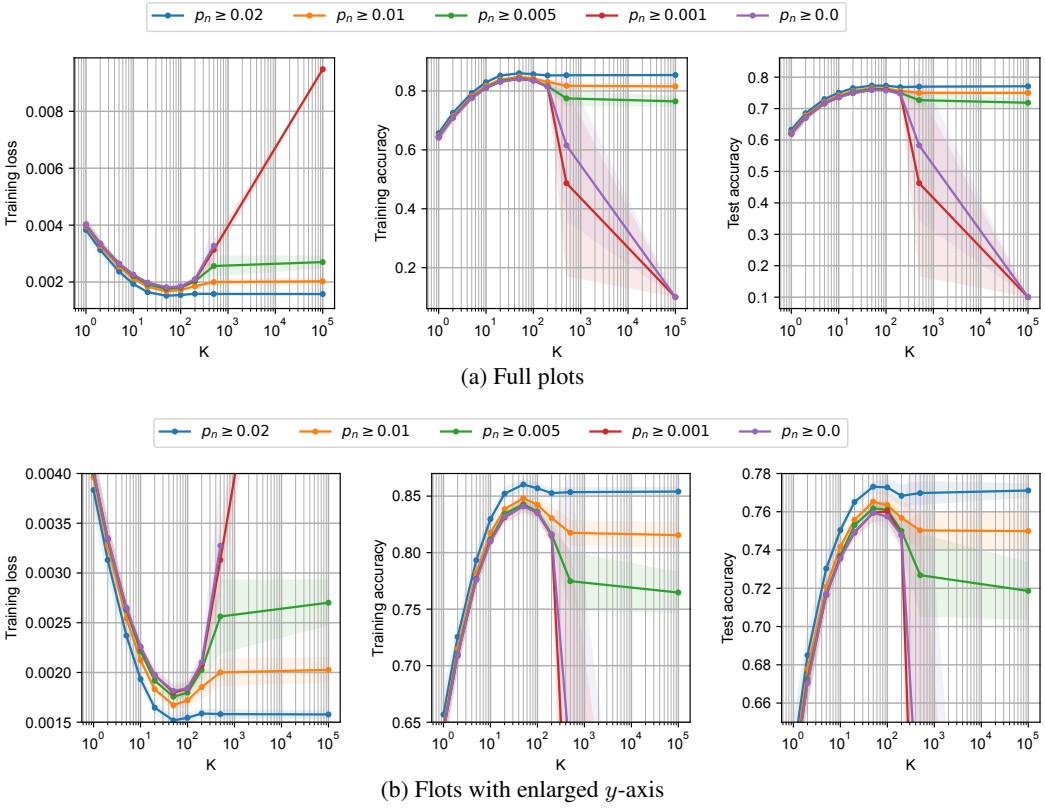

Figure D.5.2: Results of FedAU with different $K$, on CIFAR-10 dataset with Bernoulli participation. The loss is NaN for $K = 10^5$ with $p_n \geq 0.0$.

## D.6 Low Participation Rates

To further study the performance of FedAU in the presence of clients with low participation rates, we set the lower bound of participation probabilities to $0.0$ (i.e., we do not impose a specific lower bound; see Appendix C.6.1 and Appendix D.5 for details) and compare the performance of FedAU with $K = 50$ to the baseline algorithms. We consider settings with different mean participation probabilities $\mathbb{E}[p_n]$, while following the same procedure of generating heterogeneous participation patterns as described in Appendix C.6, to capture the effect of different *overall* participation rates of clients. The resulting distributions of $\{p_n\}$ with different $\mathbb{E}[p_n]$ are shown in Figure D.6.1.

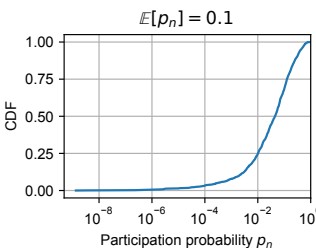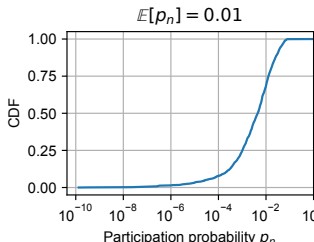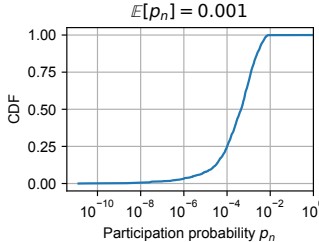

Figure D.6.1: Distributions of participation probabilities $\{p_n\}$ at clients, with different values of $\mathbb{E}[p_n]$. The distribution is expressed as empirical cumulative distribution function (CDF) with logarithmic scale on the $x$-axis.

Table D.6.1: Accuracy results (in %) on training and test data of CIFAR-10, with different mean participation probabilities $\mathbb{E}[p_n]$ of clients (Bernoulli participation) and no minimum cap of $p_n$

| **Mean participation probability** | $\mathbb{E}[p_n] = 0.1$ | | $\mathbb{E}[p_n] = 0.01$ | | $\mathbb{E}[p_n] = 0.001$ | |
|---|---|---|---|---|---|---|
| **Method / Metric** | Train | Test | Train | Test | Train | Test |
| FedAU (ours, $K = 50$) | **84.1**±0.6 | **75.9**±0.5 | **71.5**±1.3 | **67.7**±1.1 | **45.9**±1.2 | **45.6**±1.2 |
| Average participating | 81.6±0.7 | 72.5±0.7 | 60.2±1.4 | 57.9±1.5 | 26.7±1.8 | 26.8±1.8 |
| Average all | 79.5±0.8 | 71.5±0.9 | 61.8±2.4 | 60.0±2.5 | 33.0±1.8 | 33.5±1.9 |
| FedVarp (250× memory) | 61.5±25.8 | 59.5±24.8 | 10.0±0.0 | 10.0±0.0 | 12.7±3.4 | 12.8±3.5 |
| MIFA (250× memory) | 74.8±1.9 | 72.5±1.5 | 10.0±0.0 | 10.0±0.0 | 10.0±0.0 | 10.0±0.0 |
| Known participation statistics | 15.0±10.0 | 14.9±9.8 | 10.0±0.0 | 10.0±0.0 | 10.0±0.0 | 10.0±0.0 |

*Note to the table.* Total number of rounds is $10,000$. We do not enforce a minimum participation probability in these results, i.e., we allow any $p_n \geq 0.0$. We use the hyperparameters listed in Table C.4.1 for all the experiments. The same note in Table 1 also applies to this table.

**Key Observations.** The accuracy results are presented in Table D.6.1, from experiments with the CIFAR-10 dataset and $10,000$ rounds of FL. As expected, the performance of the majority of algorithms decreases as $\mathbb{E}[p_n]$ decreases, where the minor increase of FedVarp's performance from the case of $\mathbb{E}[p_n] = 0.01$ to $\mathbb{E}[p_n] = 0.001$ is due to randomness in the experiments. We summarize the key findings from Table D.6.1 in the following.

It is interesting to see that the baseline algorithms that require additional memory or other information actually perform very poorly when the clients' participation rates are low, where we note that an accuracy of $10\%$ corresponds to random guess for the CIFAR-10 dataset that has 10 classes of images. The reason is that FedVarp and MIFA both perform variance reduction based on previous updates of clients. When clients participate rarely, it is likely that the saved updates are outdated, causing more distortion than benefit to parameter updates. For the case of known participation statistics, the aggregation weight of each client $n$ is chosen as $1/p_n$. When $p_n$ is very small, the aggregation weight becomes very large, which causes instability to the model training process.

The average participating and average all baselines perform better than the FedVarp, MIFA, and known participation statistics baselines, because the aggregation weights used by the average participating

and average all algorithms do not have much variation, which provides more stability in the case of low client participation rates.

FedAU (with $K = 50$) gives the best performance, because the cutoff interval of length $K$ ensures that the aggregation weights are not too large, which provides stability in the training process. At the same time, clients that participate frequently still have lower aggregation weights, which balances the contributions of clients with different participation rates.

**Further Discussion.** We further note that all the results in Table D.6.1 are from experiments using the hyperparameters listed in Table C.4.1. These near-optimal hyperparameters were found from a grid search (see Appendix C.4) when the clients participate according to Bernoulli distribution with statistics described in Appendix C.6. For the baseline methods that give random guess (or close to random guess) accuracies, it is possible that their performance can be slightly improved by choosing a much smaller learning rate, which may alleviate the impact of stale updates (for FedVarp and MIFA) or excessively large aggregation weights (for known participation statistics) when the client participation rate is low. However, it is impractical to fine tune the learning rates depending on the participation rates, especially when the participation rates are unknown a priori. In addition, using very small learning rates generally slows down the convergence, although the algorithm may converge in the end after a large number of rounds. The fact that FedAU gives the best performance compared to the baselines for a wide range of client participation rates, while keeping the learning rates unchanged (i.e., using the values in Table C.4.1), confirms its stability and usefulness in practice.

