# OpenReview forum: "A Lightweight Method for Tackling Unknown Participation Statistics in Federated Averaging"
_ICLR.cc/2024/Conference — ICLR 2024 spotlight_

### Official Review · Reviewer_ByHL · 2023-10-22

**Soundness:** 3 good
**Presentation:** 3 good
**Contribution:** 3 good
**Rating:** 6
**Confidence:** 4

**Summary:**

The paper addresses a critical issue in federated learning (FL) where clients have varying and unknown participation rates, which can hinder FL's performance. Existing solutions often rely on global variance reduction, which consumes substantial memory resources. The paper introduces a lightweight method called FedAU, which adapts aggregation weights in FedAvg based on each client's participation history. FedAU resolves this problem by adaptively weighting client updates using online estimates of optimal weights, even without knowledge of participation statistics. Theoretical analysis shows that FedAU converges to the original objective's optimal solution with desirable properties such as linear speedup, and experimental results support its advantages over baseline methods in various participation scenarios.

**Strengths:**

The strengths of this paper's contributions are as follows:

1. The authors introduce a lightweight procedure named FedAU for estimating optimal aggregation weights for each client based on their participation history. This approach supports FL even when participation statistics are unknown, making it highly practical.

2. The paper provides a novel and thorough analysis of the convergence upper bound for FedAU. It employs a unique method to handle weight error in the convergence bound and shows that FedAU converges to the optimal solution of the original objective. Furthermore, it achieves desirable linear speedup in convergence when the number of FL rounds is sufficiently large.

3. Experimental results validate the advantages of FedAU over various datasets and baseline methods, particularly in scenarios with diverse participation patterns, including independent, Markovian, and cyclic patterns. This demonstrates the robustness and effectiveness of the proposed approach.

**Weaknesses:**

1. The theoretical results in the paper are founded on the assumption of a Bernoulli distribution for client participation. It does introduce an extra layer of specificity that might not hold universally. It would be beneficial for the authors to explicitly state this assumption within the paper, as the current presentation of the four assumptions largely omits this fact.

2. I wonder if there is a potential limitation in scenarios where clients rarely participate in the training process, as seen in cases like cross-device federated learning. In such instances, the online estimation of the aggregation weights (w_t) could be challenging or, at best, very inaccurate.

**Questions:**

see above.

---

> ### Author Response · Authors · 2023-11-17
> **Response to Reviewer ByHL**
>
> Many thanks for your review and the positive rating. We have revised the paper to address the weaknesses mentioned in your review. The main changes are highlighted in blue text in the paper. Our response is as follows.
>
> **Explicitly state the assumption on Bernoulli distribution in the theoretical analysis**
>
> In our original submission, we stated in Assumption 4 that each client $n$'s participation instance $\mathbb{I}_t^n$ is independent across $t$ and its mean is $p_n$. This implies a Bernoulli distribution with probability $p_n$. To make it clear, in the revised paper, we have added a mentioning to Assumption 4 saying that $\mathbb{I}_t^n$ follows a Bernoulli distribution with probability $p_n$.
>
> Regarding this theoretical limitation, we have explained why it is challenging to analyze the more general setting in the paragraph with the heading "Challenge in Analyzing Time-Dependent Participation" on page 6. We restate it here for clarity: Regarding the assumption of independence of $\mathbb{I}_t^n$ across time (round) $t$ in Assumption 4, the challenge in analyzing the more general time-dependent participation is due to the complex interplay between the randomness in stochastic gradient noise, participation identities $\{\mathbb{I}_t^n\}$, and estimated aggregation weights $\{\omega_t^n\}$. In particular, the first step in our proof of the general descent lemma (see Appendix B.3, the specific step is in (B.3.6)) would not hold if $\mathbb{I}_t^n$ is dependent on the past, because the past information is contained in $\mathbb{x}_t$ and $\{\omega_t^n\}$ that are conditions of the expectation. We emphasize that this is a *purely theoretical limitation*, and this time-independence of client participation has been assumed in the majority of works on FL with client sampling (Fraboni et al., 2021a;b; Karimireddy et al., 2020; Li et al., 2020b;c; Yang et al., 2021). The novelty in our analysis is that we consider the true values of $\{p_n\}$ to be *unknown* to the system. Our experimental results in Section 5 show that FedAU provides performance gains also for Markovian and cyclic participation patterns that are both time-dependent.
>
>
> **Clients rarely participate in the training process**
>
> We have added new results and discussion to Appendix C.11, to study the performance when clients participation rarely. The key conclusion is that FedAU gives the best performance compared to the baselines for a wide range of client participation rates, which confirms its stability and usefulness in practice.
>
> Although the error of aggregation weight estimation increases as the participation rate decreases, the uniqueness of our FedAU algorithm is that it includes a cutoff interval of length $K$ when we perform the aggregation weight estimation in Algorithm 2. Intuitively, due to this cutoff interval, FedAU tends to use a fixed aggregation weight when the client participates less frequently, and it tends to use an adaptive aggregation weight when the client participates more frequently. This parameter $K$ strikes a nice balance between stability (for clients with low participation rates and high estimation error) and adaptivity (for clients with higher participation rates). For this reason, as we see in the newly added Appendix C.11, although the performance of FedAU decreases as the participation rate decreases, we do not see a cutoff point where the algorithm does not work at all. This is in contrast to the FedVarp, MIFA, and even the known participation statistics baselines as shown in Table C.11.1, where the algorithms basically stop working at low participation rates due to the instability caused by the nature of those algorithms. See Appendix C.11 for more explanation, and also Appendix C.10 for results on the effect of $K$.
>
>
> Thanks again!

---

> > ### Comment · Reviewer_ByHL · 2023-11-20
> >
> > Thanks for the authors' prompt response. I will discuss it with AC as well as other reviewers in the later phases. For now, I do not have any further questions.

---

> ### Author Response · Authors · 2023-11-21
> **Thanks for the response**
>
> Dear Reviewer ByHL,
>
> Thanks a lot for your confirmation! We appreciate it.
>
> Best regards,
> Authors of Paper 4628

---

### Official Review · Reviewer_X6kq · 2023-11-01

**Soundness:** 3 good
**Presentation:** 3 good
**Contribution:** 3 good
**Rating:** 5
**Confidence:** 4

**Summary:**

This paper studies the problem of partial participation in federated learning. More specifically, the authors consider the FedAvg algorithm and assume that each client $m$ will participate with some unknow probability $p_m$. The authors show that if we are aiming to optimize the objective in equation (1), the non-adaptive aggregation weight in FedAvg will lead to a solution of optimizing another objective. In addition, the authors propose a new method that can compute the adaptive aggregation weight efficiently, and provide the corresponding convergence rate.

**Strengths:**

The strengths of the paper:
1. The authors provide a result to show that non-adaptive aggregation weight in FedAvg is bad for optimizing the objective in equation (1).
2. The authors develop an efficient method to estimate the adaptive aggregation weight that can be used in FedAvg.
3. The authors establish the convergence rate of the proposed method which demonstrates the effectiveness of the proposed method.

**Weaknesses:**

The weaknesses of the paper:
1. There is no (theoretical) comparison with existing baselines.
2. Several conditions in the established results are unclear.

**Questions:**

I have the following questions regarding the current paper:
1. What are the other baseline algorithms and their corresponding convergence rates?
2. According to Theorem 1, when we have full participation, i.e., $p_n=1$ and $w_n=1$, the objective in (2) will reduce to the objective in (1). Therefore, whether your results will recover the convergence rate of FedAvg with full participation? In addition, when we have partial participation with $p_n=p<1$ and $w_n=1$, the objective in (2) will also reduce to the objective in (1). How will your result look like compared to the existing results?
3. What is the expression on $\Psi_G$ in your theorems?
4. In Corollary 4, what do you mean by sufficient large $T$? Do you mean the results hold only asymptotically?

---

> ### Author Response · Authors · 2023-11-17
> **Response to Reviewer X6kq**
>
> Many thanks for your review. We address your questions in the following. We have also made related edits to the paper and the main changes are highlighted in blue text in the paper.
>
> **Other baseline algorithms and convergence rates**
>
> We answer both Questions 1 and 2 here.
>
> We have added Appendix A.3 to discuss the comparison with existing convergence results for FedAvg with **partial participation**. *Please see Appendix A.3 for details.* In short, when the participation probabilities are both homogeneous and known, the result in Corollary 4 of our paper **recovers** the FedAvg convergence bound of $\mathcal{O}\left(\frac{1}{\sqrt{ST}}\right)$ for non-convex objectives in Theorem 1 in [a] and improves over the bound of $\mathcal{O}\left(\frac{\sqrt{I}}{\sqrt{ST}}\right)$ in Corollary 2 in [b], where $S$ is the number of clients that participate in each round and $I$ is the number of local updates.
>
> Regarding FedAvg with **full participation**, we emphasize that **even existing convergence results for the original FedAvg algorithm**, such as those in [a] and [b], **are unable to exactly match** the convergence rates for partial participation with that for full participation. In the case of full participation, a convergence bound of $\mathcal{O}\left(\frac{1}{\sqrt{SIT}}\right)$ can be obtained (Theorem 1 in [a] with $S=N$ and Corollary 1 in [b]) for SGD with $\sigma>0$. The difference in the coefficient $\sqrt{I}$ in the denominator is due to a technical step in the convergence proof for the original FedAvg algorithm. It is also referred to as the partial participation error in Corollary 1 in [c] (the result of Corollary 1 in [c] is order-wise the same as Corollary 2 in [b]). However, if the number of local updates $I$ is set as a constant that is absorbed in $\mathcal{O}(\cdot)$, the results for both partial and full participation become the same (order-wise), giving $\mathcal{O}\left(\frac{1}{\sqrt{ST}}\right)$.
>
> For other baselines that are based on **global variance reduction**, including FedVarp and MIFA, it is difficult to make a fair comparison because they require a substantial amount of additional memory that is equal to the size of a model *for every client*. For example, the CNN model that we use in our experiments for the CIFAR-10 dataset has $551,786$ parameters. In contrast, our FedAU algorithm only needs to save three additional scalar values per client, as explained in Section 3 after we describe Algorithm 2. This means that these baselines incur $\frac{551,786}{3}-1 \approx 183,928$ times of *additional* memory overhead compared to our FedAU approach, and the overhead is even larger for bigger models. Note that, here, we are *not* counting the memory for keeping a single model, which is needed for all algorithms including the original FedAvg, and even for inference tasks without training; thus, the number here is different from the $250\times$ that we included in Table 1. It is difficult to incorporate memory consumption in theoretical convergence analysis. Regardless of their substantial amount of additional memory consumption, these baselines have a convergence bound of $\mathcal{O}\left(\frac{1}{\sqrt{SIT}}\right)$ for SGD with $\sigma>0$, which is the same as FedAvg with full client participation, and it also becomes $\mathcal{O}\left(\frac{1}{\sqrt{ST}}\right)$ if $I$ is a constant absorbed in $\mathcal{O}(\cdot)$. Moreover, the practical performance of algorithms based on global variance reduction is often not very good. In Section 5 of our paper and also in Appendix C.11 that we have added in the revision, we have experimentally verified the advantage of our FedAU algorithm compared to these baselines, for several datasets, participation patterns, and participation rates.
>
> [a] SCAFFOLD: Stochastic Controlled Averaging for Federated Learning, ICML 2020.
> [b] Achieving Linear Speedup with Partial Worker Participation in Non-IID Federated Learning, ICLR 2021.
> [c] FedVARP: Tackling the Variance Due to Partial Client Participation in Federated Learning, UAI 2022.
>
> **Expression on $\Psi_G$**
>
> The variable $\Psi_G$ has already been defined in Theorem 2 in our original submission. Depending on which additional assumption is used, the value of $\Psi_G$ is either zero or $G$. In the revised paper, we have swapped the order of presentation in Theorem 2 and put the definition of $\Psi_G$ before Equation (6), so that it appears more prominently to the reader.
>
> **Meaning of sufficiently large $T$ in Corollary 4**
>
> In our original submission, the notion of sufficiently large $T$ means that $T$ is large enough so that certain conditions are met, but we do *not* require $T\rightarrow\infty$. We realize that this notion is implicit. In the revised paper, we have updated the result in Corollary 4 and its proof. **Our updated result holds for any $T\geq 2$** and therefore we have removed the mentioning of "sufficiently large $T$".
>
> Thanks again!

---

> > ### Author Response · Authors · 2023-11-22
> > **Discussion Reminder**
> >
> > Dear Reviewer X6kq,
> >
> > We hope that our response has addressed your concerns. As today is the last day of the reviewer-author discussion phase, please let us know if you have any further question. Thanks a lot!
> >
> > Best regards,
> > Authors of Paper 4628

---

### Official Review · Reviewer_p8tC · 2023-11-08

**Soundness:** 3 good
**Presentation:** 3 good
**Contribution:** 3 good
**Rating:** 6
**Confidence:** 4

**Summary:**

This paper studies the FedAvg algorithm with unknown participation statistics. It first proves that FedAvg with non-optimal aggregation weights can diverge from the optimal solution of the original FL objective. Next, it proposes an adaptive method to estimate the participation weight and come up with the FedAU algorithm that can converge to the desired solution even under the unknown participation statistics. Numerical experiments validate the theoretical findings.

**Strengths:**

1. It shows that, with unknown participation statistics, FedAvg with non-optimal weight will diverge from the optimal solution of the original FL objective.

2. It proposes an interesting online Algorithm 2 to estimate the unknown participation weight.

3. It proposes FedAU algorithm that can converge to the desired solution even with unknown participation weight.

**Weaknesses:**

The linear speedup term in Eq.(8) seems unreasonable. According to the FedAvg result in reference [R1] (see Table 2), when only a subset of clients (say S clients) participate in the FedAvg, the linear speedup term should be O(1/sqrt{S*I*T}), not O(1/sqrt{N*I*T}).  I believe O(1/sqrt{S*I*T}) makes more sense since only S clients sample data and participate in algorithm update per iteration.

[R1] Karimireddy et.al., SCAFFOLD: Stochastic Controlled Averaging for Federated Learning, ICML 2020.

**Questions:**

1. Please clarify why does your linear speedup term is O(1/sqrt{N*I*T}) not O(1/sqrt{S*I*T}) as shown in reference [R1] with partial client sampling. I think O(1/sqrt{S*I*T}) makes more sense since only a subset of S participates in data sampling and model update per iteration.

2. Why do you need both bounded variance assumption in (4) and bounded global gradient assumption in Theorem 2? The bounded gradient assumption is typically very restrictive and are not used in literature such as in [R1]. Can the bounded global gradient assumption be removed?

---

> ### Author Response · Authors · 2023-11-17
> **Response to Reviewer p8tC**
>
> Many thanks for your review. We would like to point out that the concerns are mostly due to misunderstanding. We clarify them in the following and have also added some related discussions to the paper (the main changes are in blue text).
>
>
> **Clarification on linear speedup term**
>
> We note that our convergence bound includes the parameter $Q$ that is defined as $Q:= \max_{t\in\{0,\ldots,T-1\}} \frac{1}{N}\sum_{n=1}^N p_n (\omega_t^n)^2$ in Theorem 2. When we know the participation probabilities, as in [R1], and choose $\omega_t^n=\frac{1}{p_n}$ for all $t$ we have $Q=\frac{1}{N}\sum_{n=1}^N \frac{1}{p_n}$. Further, for equiprobable sampling of $S$ clients out of a total of $N$ clients, which is the setup considered in [R1], we have $p_n = \frac{S}{N}$ and thus $Q=\frac{N}{S}$. When $T$ is large and ignoring the other constants, our upper bound in Corollary 4 becomes $\mathcal{O}\left(\frac{\sqrt{Q}}{\sqrt{NT}}\right)=\mathcal{O}\left(\frac{1}{\sqrt{ST}}\right)$, where the linear speedup is with respect to $S$ in this special case.
>
> This result is the same as Theorem 1 in [R1] for non-convex objectives. We also note that, in the result in [R1], the dominating term does *not* include the number of local update steps $I$ in the denominator when $S<N$. See the definition of $M$ in the first footnote in Table 2 of [R1] and also see the last equation in Theorem 1 of [R1]. This is due to a technical step needed in the proof of convergence of the original FedAvg algorithm.
>
> The uniqueness of our work compared to [R1] and most other existing works is that we consider heterogeneous and unknown participation statistics (probabilities), where each client $n$ has its own participation probability $p_n$ that can be different from other clients. In contrast, [R1] assumes uniformly sampled clients where a fixed (and known) number of $S$ clients participate in each round. Therefore, our setup is more general where the number of clients that participate in each round can vary over time, and we cannot define a fixed value of $S$ in this general setup. Because of this generality, we use $Q$ to capture the statistical characteristics of client participation. When the overall probability distribution of client participation remains the same, increasing the total number of clients has the same effect as increasing the number of participating clients, as we have shown above.
>
> See the newly added Appendix A.3 for more details.
>
>
> **Regarding bounded global gradient assumption**
>
> First, we note that the assumption on bounded variance of stochastic gradients (Equation (4) in Assumption 2) is applied to each *local* objective $F_n(\mathbf{x})$, whereas the bounded global gradient assumption is applied to the *global* objective $f(\mathbf{x})$ that is defined as an average of the local objectives. These two assumptions do not imply each other, since one is about the variance of the *stochastic* gradient of the local objective and the other is about the norm of the *full* gradient of the global objective.
>
> Second, we emphasize that, as stated in Theorem 2, our convergence result holds when **either** of the "bounded global gradient" assumption or the "nearly optimal weights" assumption holds. When the aggregation weights $\{\omega_t^n\}$ are nearly optimal satisfying $\frac{1}{N}\sum_{n=1}^N \left(p_n \omega_t^n - 1\right)^2 \leq \frac{1}{81}$, we do not need the bounded gradient assumption.
>
> For the bounded gradient assumption itself, as we already mentioned in the paragraph after Theorem 2, a stronger assumption of bounded stochastic gradient is used in related works on adaptive gradient algorithms (Reddi et al., 2021; Wang et al., 2022b;c - references in the paper), which implies an upper bound on the per-sample gradient. Compared to these works, we only require an upper bound on the global gradient, i.e., average of per-sample gradients, in our work. Although focusing on very different problems, our FedAU method shares some similarities with adaptive gradient methods in the sense that we both adapt the weights used in model updates, where the adaptation is dependent on some parameters that progressively change during the training process. The difference, however, is that our weight adaptation is based on each client's participation history, while adaptive gradient methods adapt the element-wise weights based on the historical model update vector. Nevertheless, the similarity in both methods leads to a technical (mathematical) step of bounding a "weight error" in the proofs, which is where the bounded gradient assumption is needed especially when the "weight error" itself cannot be bounded. In our work, this step is done in the proof of Theorem 2 (in Appendix B.5). In adaptive gradient methods, as an example, this step is on page 14 until Equation (4) in Reddi et al., 2021 (reference in the paper).
>
> We have added a discussion related to this in Appendix A.2. See Appendix A.2 for more details.
>
> Thanks again!

---

> > ### Comment · Reviewer_p8tC · 2023-11-21
> > **Thanks for the rebuttal**
> >
> > Many thanks for the detailed rebuttal. My concern on the linear speedup has been addressed. Please include your discussions on linear speedup in terms of the $S$ in the revised version.
> >
> > As to the global gradient assumption, I still believe it is a strong assumption since the standard assumption should be the bounded variance of each stochastic gradient and the bounded gradient dissimilarity as used in [R1], but not bounded global gradient. While I agree that some adaptive FL algorithms like FedAdam use bounded gradient, I don't think your algorithm, while adaptive in estimating participation weight, belong to the adaptive stochastic gradient family. The bounded gradient assumption used in FedAdam does not imply the necessity of such assumption in your algorithm.
> >
> > Except for this point, I think the paper is novel and interesting in general. I will increase my score to 6.

---

> > > ### Author Response · Authors · 2023-11-21
> > > **Thanks for the response**
> > >
> > > Dear Reviewer p8tC,
> > >
> > > Many thanks for your response and increasing the score! We have expanded Appendix A.3 to include the discussion related to linear speedup. The revised paper has been uploaded.
> > >
> > > Regarding the assumption on bounded global gradient, we note that FedAdam uses a much stronger assumption on an *element-wise* bound of the *per-sample* gradient (see Assumption 3 in Reddi et al., 2021, Adaptive Federated Optimization). Our assumption on bounded *global* gradient is much weaker than theirs.
> > >
> > > The reason that we need either the “nearly optimal weights” assumption or the “bounded global gradient” assumption is the term $$\gamma\eta I  \left[\frac{81}{16N}\sum\_{n=1}^N \left(p\_n \omega\_t^n - 1\right)^2 + 15L^2I^2\gamma^2 + \frac{27\gamma\eta LI}{16N^2}\sum\_{n=1}^N p\_n (\omega\_t^n)^2  - \frac{1}{4} \right] \cdot\\|\nabla f(\mathbf{x}\_{t})\\|^2$$ in the right-hand side (RHS) of our descent lemma (Lemma B.5.1 on page 24 in the appendix). For the final result to hold, either $\frac{81}{16N}\sum_{n=1}^N \left(p_n \omega_t^n - 1\right)^2$ needs to be small enough so that the whole expression above remains bounded by a negative constant (note the existence of the quantity $- \frac{1}{4}$ in the above expression), or the global gradient norm $\\|\nabla f(\mathbf{x}_{t})\\|^2$ needs to be bounded, which corresponds to the “nearly optimal weights” assumption or the “bounded global gradient” assumption, respectively. The reason for this requirement is  apparent from the subsequent steps in the proof of Theorem 2 (starting near the end of page 24 in the appendix).
> > >
> > > The quantity $\frac{81}{16N}\sum_{n=1}^N \left(p_n \omega_t^n - 1\right)^2$ in the above expression, which is directly related to the weight error term that we mention in the paper, is due to the fact that the participation probabilities $\\{p_n\\}$ are unknown and we need to estimate the aggregation weights $\\{\omega_t^n\\}$, which is unique to our work. This quantity is also the fundamental cause of requiring either of these two additional assumptions. It first appears in a slightly different form in Equations (B.3.7) and (B.3.8) (on page 20-21 in the appendix), when we bound the progression of the objective function value by applying the property of $L$-smoothness.
> > >
> > > If the reviewer has any thoughts on how to prove the result without the bound global gradient assumption, we would be delighted to hear about it.
> > >
> > > Again, thank you very much!
> > >
> > > Best regards,
> > > Authors of Paper 4628

---

### Official Review · Reviewer_dwXt · 2023-11-09

**Soundness:** 3 good
**Presentation:** 3 good
**Contribution:** 3 good
**Rating:** 10
**Confidence:** 3

**Summary:**

The article tackles the problem of Federated Learning under heterogeneous client participation. The authors propose a correction of FedAvg algorithm which handles heterogeneous participation through estimation of optimal aggregation weights based on client participation history, in a new algorithm called FedAU. Theoretical results are derived, which highlight sub-optimality of classical FedAvg algorithm in this setting, as well as convergence analysis of FedAU. Numerical experiments illustrate the theoretical findings.

I have read the authors response, which thoroughly answered my questions. In this regard, I think the paper is very strong and updated my score accordingmy.

**Strengths:**

- The paper tackles an important practical limitation on existing FL algorithms, which often depend on a known, homogeneous client's participation rate.
- The proposed algorithm is original as it tackles client participation heterogeneity using novel methodologies quite different from existing litterature mostly based on variance reduction techniques. In addition, the proposed algorithm enjoys favourable computational complexity compared to existing works.
- The proof techniques are also original and could be reused in other settings
- The paper is very clear and well-written, the problematic, related work and contributions are clearly highlighted, and scientific methodology is easy to follow. In particular, the authors made the effort of stating intuitive results and presenting formal theorems in a user-friendly manner.

**Weaknesses:**

- The authors mention that the proof techniques, but I didn't find any sketch of the proof in the main text, which is a shame because it could help readers understand the novelty, and maybe reuse proof techniques.
- The numerical experiments are a bit disappointing in the sense that they do not really highlight how much of the theoretical results are observed in practice, but only that FedAU improves over FedAvg on the proposed use cases. It would have been interesting to see experiments highlighting convergence error for varying values of K or ground truth average participation rates (even on simple simulations)

**Questions:**

- In practice, what is the order of magnitude of the lowest participation rate that can be estimated?
- Related question, did you perform any experiments to understand at what point the weights might explode (in the case where K is very large) ?
- It is also likely that clients participation rate pn actually vary over time, while remaining independent across t. Do you think your theoretical results could easily be adapted to such cases ?
- Concerning the theoretical results, using your proof techniques, do you recover existing results in the particular case where pn are homogeneous/known ?

---

> ### Author Response · Authors · 2023-11-17
> **Response to Reviewer dwXt**
>
> Many thanks for your review and the positive rating. We have revised the paper to address the weaknesses and questions mentioned in your review. The main changes are highlighted in blue text in the paper. Our response is as follows.
>
>
> **Proof sketch in the main text**
>
> We have added sentences after Theorems 2 and 3 and Corollary 4, to highlight the unique steps in our proofs.
>
>
> **Numerical experiments on how much of the theoretical results are observed in practice**
>
> We ran new experiments for both different values of $K$ and different client participation rates. The new results and discussions have been added to Appendix C.10 and Appendix C.11. Please see these two sections in the appendix for details.
>
>
> **In practice, what is the order of magnitude of the lowest participation rate that can be estimated?**
>
> In general, the estimation error increases as the participation rate decreases, because the number of participation intervals that are collected as samples for such estimation becomes less. However, the uniqueness of our FedAU algorithm is that it includes a cutoff interval of length $K$ when we perform the aggregation weight estimation in Algorithm 2. Intuitively, due to this cutoff interval, FedAU tends to use a fixed aggregation weight when the client participates less frequently, and it tends to use an adaptive aggregation weight when the client participates more frequently. This parameter $K$ strikes a nice balance between stability (for clients with low participation rates and high estimation error) and adaptivity (for clients with higher participation rates). For this reason, as we see in the newly added Appendix C.11, although the performance of FedAU decreases as the participation rate decreases, we do not see a cutoff point where the algorithm does not work at all. This is in contrast to the FedVarp, MIFA, and even the known participation statistics baselines as shown in Table C.11.1, where the algorithms basically stop working at low participation rates due to the instability caused by the nature of those algorithms (see the Appendix C.11 for more explanation).
>
>
> **Did you perform any experiments to understand at what point the weights might explode (in the case where K is very large)?**
>
> The experimental results of FedAU with different $K$ have been added in Appendix C.10. The results show that, when there exists a subset of clients with very low participation rates, choosing $K\geq 500$ causes a significant performance drop, i.e., the weights explode in this case. However, the performance remains similar when we choose a smaller $K$, where $K=50$ gives the best performance. This further confirms the advantage and necessity of having a cutoff interval of a finite length $K$, which is a unique component in our FedAU algorithm.
>
>
> **Can the theoretical results be adapted to the case of $p_n$ varying over time, while remaining independent across $t$?**
>
> We believe it is possible to extend the convergence result in Theorem 2 to time-varying $p_n$, as long as the probabilities are given for all $t$ and their corresponding random instances of participation are independent across $t$. The result will be very similar to that in Theorem 2, where $p_n$ would include an additional subscript $t$. However, it is difficult to obtain results that are similar to those in Theorem 3 and Corollary 4, because the proof of Theorem 3 starting at the bottom of page 27 in the appendix considers the same $p_n$ across time. Intuitively, the term $\mathbb{E}[\left(p_n \omega_t^n - 1\right)^2]$ characterizes the estimation error, and it is generally impossible to estimate a stastistical variable related to $p_n$ if $p_n$ arbitrarily changes over time. It may be possible to bound the estimation error if we impose further assumptions on how $p_n$ varies over time, but such kinds of assumptions may also be deemed unrealistic. Due to these difficulties, we do not make such an extension in the theoretical analysis of this work. Note that our experiments in Section 5 consider more general types of participation patterns including those that are Markovian or cyclic.
>
>
> **Do you recover existing theoretical results in the particular case where $p_n$ are homogeneous/known?**
>
> We have added Appendix A.3 which answers this question. Please see Appendix A.3 for details. In short, when the participation probabilities are both homogeneous and known, the result in Corollary 4 of our paper recovers the FedAvg convergence result in Theorem 1 in [a] and improves over the result in Corollary 2 in [b].
>
> [a] SCAFFOLD: Stochastic Controlled Averaging for Federated Learning, ICML 2020.
> [b] Achieving Linear Speedup with Partial Worker Participation in Non-IID Federated Learning, ICLR 2021.
>
> Thanks again!

---

> > ### Author Response · Authors · 2023-11-22
> > **Discussion Reminder**
> >
> > Dear Reviewer dwXt,
> >
> > We hope that our response has addressed your questions and comments. As today is the last day of the reviewer-author discussion phase, please let us know if you have any further question. Thanks a lot!
> >
> > Best regards,
> > Authors of Paper 4628

---

> > ### Comment · Reviewer_dwXt · 2023-11-23
> >
> > Many thanks for your thorough answers, and congratulations on a very good paper. I have updated my score accordingly.

---

### Author Response · Authors · 2023-11-20

Dear Reviewers,

Thank you again for your time and effort in reviewing our paper! We have responded to your comments in the threads below and also made edits to the paper. As the reviewer-author discussion period ends in 2 days, we are eager to hear your feedback on whether our answers and updates are satisfactory. If you have more questions, please let us know and we will be happy to answer them. If not, we would greatly appreciate if you could kindly consider upgrading your score to reflect that the concerns have been addressed.

Thanks a lot!

Best regards,
Authors of Paper 4628

---

### Meta-Review · Area_Chair_31zm · 2023-12-17

**Metareview:**

In federated learning different clients may have different participation rates. This paper shows that in this case simple unweighted aggregation can be erroneous; the key algorithmic novelty in this paper is to propose a new weighting scheme. This scheme weighs down rarely participating clients in a specific way. Theoretical guarantees are established for this procedure.

**Justification For Why Not Higher Score:**

The paper's key idea is intuitive, the innovation is mainly in the analysis. Scores don't justify an oral.

**Justification For Why Not Lower Score:**

Strong endorsements from a couple reviewers that are well justified. the problem is well motivated enough that putting a spotlight on the solution would help.

---

### Decision · Program_Chairs · 2024-01-16

Accept (spotlight)